# A Systematic Assessment of Drought Termination in the United Kingdom

**S. Parry[1,2], R. L. Wilby[2], C. Prudhomme[1,2] and P. J. Wood[2]**

[1]{Centre for Ecology & Hydrology, Wallingford, Oxfordshire, OX10 8BB, UK}

[2]{Department of Geography, Loughborough University, Loughborough, Leicestershire, LE11 3TU, UK}

Correspondence to: S. Parry (spar@ceh.ac.uk)

## Abstract

Drought termination can be associated with dramatic transitions from drought to flooding. Greater attention may be given to these newsworthy and memorable events, but drought terminations that proceed gradually also pose challenges for water resource managers. This paper defines drought termination as a distinctive phase of the event. Using observed river flow records for 52 UK catchments, a more systematic and objective approach for detecting drought terminations is demonstrated. The parameters of the approach are informed by a sensitivity analysis that ensures a focus on terminations of multi-season to multi-year droughts. The resulting inventory of 459 drought terminations provides an unprecedented historical perspective on this phenomenon in the UK. Nationally- and regionally-coherent drought termination events are identifiable, although their characteristics vary both between and within major episodes. Contrasting drought termination events in 1995-98 and 2009-12 are examined in greater depth. The data are also used to assess potential linkages between metrics of drought termination and catchment properties. The duration of drought termination is moderately negatively correlated with elevation ($r_s$=-0.48) and catchment average rainfall ($r_s$=-0.40), suggesting that wetter catchments in upland areas of the UK tend to experience shorter drought terminations. More urbanised catchments tend to have gradual drought terminations (contrary to expectations of flashy hydrological response in such areas) although this may also reflect the type of catchments typical of lowland England. Significant correlations are found between the duration of the drought development phase and both the duration ($r_s$=-0.30) and rate ($r_s$=0.28)

of drought termination. This suggests that prolonged drought development phases tend to be followed by shorter and more abrupt drought terminations. The inventory helps to place individual events within a long-term context. The drought termination phase in 2009-12 was, at the time, regarded as exceptional in terms of magnitude and spatial footprint, but the Thames river flow record identifies several comparable events before 1930. The chronology could, in due course, provide a basis for exploring the complex drivers, long-term variability and impacts of drought termination events.

## 1 Introduction

Drought termination, generally defined as the end point of a drought, has been neglected in research literature relative to drought onset. Studies which address this phenomenon have focused on extreme transitions at the end of a drought (e.g. Yang et al. 2012; Ning et al. 2013), but there has been a lack of attention devoted to assessing the full range of drought termination types and characteristics. Whilst abrupt drought terminations may result in more destructive and newsworthy impacts (e.g. Webster et al. 2011; Lavers & Villarini 2013; Parry et al. 2013), gradual drought terminations are problematic for water resource managers who must reconcile public relations with continued water restrictions during wet weather.

Some studies systematically identify and characterise droughts themselves (e.g. Hisdal et al. 2001; Pfister et al. 2006; Marsh et al. 2007; Fleig et al. 2011; Li et al. 2013), but these have generally not considered the drought termination phase. A limited historical perspective can be gained from studies of drought termination on an event basis, including those based on hydrometeorological (e.g. Kienzle 2006; Marengo et al. 2008), remotely sensed (e.g. Wang et al. 2013; Chew & Small 2014) or experimental catchment data (e.g. Miller et al. 1997; Lange & Hansler 2012). Even considering several events (e.g. Eltahir & Yeh 1999; Shukla et al. 2011) is too limited a sample to generalise, or move beyond qualitative descriptions (e.g. Parry et al. 2013). A systematic assessment would enable a more robust analysis of the spatial and temporal variability of drought termination. Moreover, the importance of the end of a drought has already been recognised as a criterion in a hydrological drought typology and a basis for differentiating drought types (Van Loon & Van Lanen 2012; Van Loon et al. 2015).

Studies that systematically identify the end of droughts in the historical record (e.g. Mo 2011; Kam et al. 2013; Maxwell et al. 2013; Patterson et al. 2013) have typically considered drought termination to be instantaneous. There are two notable exceptions; Bonsal et al. (2011) sub-

divided drought into six stages, one of which is the concept of drought termination as a phase
considered herein, and Nkemdirim & Weber (1999) expressed the concept of a rate of drought
termination using Palmer Drought Severity Index units over time.
Preliminary steps have been taken to identify and characterise the spatial signature of a single
drought termination for 15 catchments in the UK (Parry et al. 2016), and to apply the same
assessment technique in a temporal analysis of drought terminations in a single catchment for
the period 1883-2013 (Parry et al. 2015).  The approach adopted in these studies differs from
others (e.g. Kam et al. 2013; Patterson et al. 2013) by considering drought termination to be a
period of a drought event with its own start, end and duration between these points.
By combining these spatial (Parry et al. 2016) and temporal approaches (Parry et al. 2015), the
aim of this study is to derive chronologies of drought termination for 52 UK catchments.  These
data are subsequently used to assess the historical variability of drought termination and to
explore the link between drought termination metrics and catchment properties.  A sensitivity
analysis of the drought termination metrics to methodological parameters is included; the
selection of parameters that results from this analysis is also informed by the focus of this study
on the termination of multi-season to multi-year droughts.  It is anticipated that a better
understanding of the physical processes driving drought termination will lead to improved
water resources management and forecasting during these problematic episodes in the future.
**2   Data**
Catchments were selected on the basis of their area and record length, favouring larger
catchments with longer records in order to maximise the spatial and temporal coverage of the
chronologies.  This selection was supplemented by additional catchments to improve
representation of the diversity of hydrogeological conditions in the UK.  The resulting 52
catchments (Fig. 1; Table A1) account for more than 40% of the gauged area of the UK whilst
capturing some of the longest river flow records.  Nearly half (21 of 52) of the catchments are
classified as near-natural, and these are predominantly located in northern and western areas of
the UK.  To the south and east and for the larger catchments, flows may be affected by
anthropogenic influences (such as abstractions and return flows) which can mask changes
associated with drought termination (Ning et al. 2013).  A naturalised river flow series is used
for the Thames; no other naturalised series are available for the study catchments.  River flow
data were obtained from the UK National River Flow Archive (NRFA).  Start dates range
between January 1883 and June 1982, but all series extend to September 2013. Time series of
monthly mean river flows were derived for each catchment for every month in which at least
90% of the daily data were available. Metadata on catchment area, median elevation, Standard-
period Average Annual Rainfall for 1961-90 (hereafter SAAR6190; Spackman 1993), Base
Flow Index (hereafter BFI; Gustard et al. 1992), and urban extent (Marsh & Hannaford 2008)
were also obtained for each catchment from the NRFA (Table A1).
**3   Methodology**
**3.1   Defining drought termination**
Drought termination is defined here as a phase of a drought, rather than an instantaneous point
in time. The threshold level method (Zelenhasić & Salvai 1987) has been applied on a monthly
time step, and drought events are sub-divided at the point of the maximum negative flow
anomaly (Bravar & Kavvas 1991) into two phases: drought development and drought
termination (Fig. 2). Drought termination is characterised by its duration (e.g. Bonsal et al.
2011), rate of change (e.g. Correia et al. 1987; Nkemdirim & Weber 1999), and seasonality
(e.g. Mo 2011).
For each catchment, monthly mean flow data were converted into a percentage anomaly of the
monthly long-term average (LTA), calculated from a 1971-2000 reference period (Eq. 1).

19          $Z_{anom\,t} = 100\,(\,(\,Z_{obs\,t}\,/\,Z_{LTA\,m}\,) - 1\,)$                    (1)

where $t$ is the time step index, $m$ is the month of the time step, $Z_{anom}$ is the percentage anomaly
at $t$, $Z_{obs}$ is the observed value at $t$, and $Z_{LTAm}$ is the LTA at $m$. Where river flow records
commence after 1971 (13 of the 52 catchments; Table A1), the monthly LTA is an average of
all available monthly mean flows within the 1971-2000 timeframe. Of these 13 catchments,
only five sets of monthly LTAs are derived from less than 24 years of available data and all
catchments have at least 19 years in the 1971-2000 period.
The start of a drought development phase ($t_{sd}$ where $s$ is start and $d$ is development; Fig. 2) is
the first month of $D$ consecutive months (pre-defined by the user) for which $Z_{anom}$ is negative.
$R$ months within the $D$-month duration are permitted to be above average, to account for minor
wet interludes during the development of the drought. Once a drought has been initiated, the
end of the drought termination phase ($t_{et}$ where $e$ is end and $t$ is termination; Fig. 2) is the last

month of $T$ consecutive months for which $Z_{anom}$ is greater than $Z_{LTAm}$. The termination magnitude (TM; Fig. 2) is $Z_{anom}$ at $t_{et}$.

The end of the drought development phase ($t_{ed}$; Fig. 2) is the month with the largest negative $Z_{anom}$ value (defining the drought magnitude, DM; Fig. 2) between $t_{sd}$ and $t_{et}$. The start of the drought termination phase ($t_{st}$; Fig. 2) is the next month after $t_{ed}$.

The conceptual diagram in Fig. 2 illustrates the two phases of drought and some of the associated drought termination metrics. The drought termination duration (DTD; Fig. 2) is the number of months between $t_{st}$ and $t_{et}$. The drought termination rate (DTR; Fig. 2) is the difference between the drought magnitude and the termination magnitude, divided by the drought termination duration. The drought termination seasonality is a code relating to the seasons through which drought termination occurs. For example, if the start of drought termination is in autumn and the end of drought termination is in the next winter, the drought termination seasonality would be 'Aut-Win'. Because seasonality is assessed on the entire drought termination period rather than its beginning or end, when drought termination durations span four or more seasons they are considered not to have a seasonality.

## 3.2  Parameter selection

At the outset, expert judgement was used to select parameters which identified well known hydrological droughts in the historical record. A drought chronology for the UK (Marsh et al. 2007) identified an average of two events per decade over the last 50 years. Experimentation with different parameter sets suggested that a moderately high value for $D$ is required to ensure a focus on multi-season and multi-year droughts. The value of $R$ must balance between identifying unrealistically large numbers of events or none at all. The hydrological variability of many catchments in the UK requires the value of $T$ to be greater than one, to account for wet interludes during droughts. Combining these findings with prior expert knowledge on drought occurrence in the UK, the following parameters were identified as appropriate for the aims of this study: $D$=10; $R$=1; $T$=2.

Once the parameters had been selected, response surfaces (e.g. Fig. 3) were used to provide quantitative support for this decision. At first glance across a range of catchment sizes, characteristics and hydroclimatic settings, the parameters above generally satisfy the approximate events per decade criteria outlined above. Two contrasting catchments were selected to illustrate typical patterns of sensitivity in the response surfaces. The Scottish Dee

(Eastern Scotland; Fig. 3, left) is a relatively wet upland catchment with impermeable geology
and a flashy hydrological response, whilst the Itchen (Southern England; Fig. 3, right) is a
relatively dry lowland catchment with permeable geology and a buffered hydrological response.
The identified combination of parameters ($D$=10; $R$=1; $T$=2) is indicated by bold boxes on the
response surfaces in Fig. 3.
The response surfaces illustrate how the numbers of drought events identified varies with
parameter selection. Fewer events were identified with increasing $D$ (moving from left to right
in Fig. 3, top left and top right) due to stricter criteria for drought initiation. Conversely,
increasing $R$ (for a given $D$ and $T$, moving from bottom to top in Fig. 3, top left and top right)
detected more events because this relaxed the initiation criteria (ratio between $D$ and $R$) to allow
more intermittent months above the average flow threshold. As $T$ increased (for a given $D$ and
$R$, moving from bottom to top in Fig. 3, top left and top right), the number of identified events
decreased as the threshold for completion of drought termination became more stringent. These
patterns were consistent across a range of catchment sizes, characteristics and hydroclimatic
settings.
Although the number of identified events was the primary verification provided by the response
surfaces, variations in the average characteristics of the resulting events were also explored.
For total drought duration (TDD), increasing $T$ for the Scottish Dee (moving from bottom to
top in Fig. 3, middle left) caused identified droughts to lengthen considerably and resulted in
merging of previously distinct events into unrealistically long periods (e.g. exceeding 120
months, or 10 years). The Itchen did not exhibit this behaviour (Fig. 3, middle right) suggesting
that individual drought events were typically separated by long spells (greater than six months)
with above threshold flows such that merging was less likely. This was consistent with the
lower variability of river flows in groundwater influenced catchments like the Itchen. Similar
contrasts between the two catchments were also apparent for drought termination rate (DTR;
Fig. 3, bottom left and bottom right), in part because duration is a component of the DTR
calculation. Higher values of $T$ caused more merging of events in responsive catchments such
as the Scottish Dee, increasing TDD (and DTD) and thereby reducing DTR. Although not
directly comparable due to the different nature of the indicators used, this finding is consistent
with a previous study of two catchments with contrasting river flow regimes in which less
stringent criteria for drought identification increased the duration of droughts in the more

responsive catchment to a far greater degree (Tallaksen et al. 1997). This suggests more stringent criteria are required for more responsive catchments.

In general, drought termination metrics showed greater sensitivity to parameter values in more responsive catchments (less responsive catchments were insensitive). Severe initiation criteria (high $D$ and low $R$) and larger values of $T$ are not appropriate for responsive catchments because these combinations are physically implausible, resulting in the merging of events into unrealistic durations with corresponding effects on derived drought termination metrics.

These key findings of the sensitivity analysis verified the initial decision on parameter selection. Values of $D$=10, $R$=1 and $T$=2 do not over- or under-represent drought occurrence for catchments of different size, geology or average rainfall, whilst primarily identifying severe multi-year and multi-season events that form the focus of this study. For these reasons the same parameter values were applied to all 52 catchments in this study, and enabled a comparison of drought termination characteristics across catchments without the influence of variations in parameter selection.

## 3.3  Correlation analysis

Potential relationships between drought termination characteristics and catchment properties were explored through a correlation analysis. Since the majority of drought termination characteristics are not normally distributed, and to limit the influence of outliers, the Spearman rank correlation test (Spearman 1944) was applied to the inventory of drought development and drought termination characteristics and catchment metadata. Correlation analysis was performed using all 52 catchments, as well as on a subset of catchments with at least 10 drought terminations events. By omitting catchments with only a few identified events, a subset of catchments is retained for which catchment average drought termination characteristics are more robust against the potential variability exhibited by individual atypical events.

## 4  Results

## 4.1  Spatio-temporal variability of drought termination

Drought termination chronologies for all 52 catchments, approximately ordered from the north-west (top) to the south-east (bottom) of the UK, are presented in Fig. 4. This allows visual

inspection of the spatial coherence of drought events over a common data period beginning in the early 1970s. At a national scale, droughts have been relatively infrequent, occurring only in 1975-77 and 1995-98. Regional droughts affected southern and eastern areas in 1988-93, 2004-07 and 2009-12. Drought-poor periods are also evident, the longest of which was the decade following the 1975-77 event during which there were few prolonged droughts at either regional or national scales.

Prior to 1970, a lack of river flow data before gauged records commenced (particularly in northern and western areas of the UK; Table A1) limits the assessment of the spatial coherence of drought phases, but events in 1962-64 and 1959 are identifiable in longer records in South-west UK, Anglian, Southern England and the Midlands. Persistent drought conditions (with intermittent drought terminations) within the 1890-1910 'Long Drought' (Marsh et al. 2007) are observed in the Thames river flow record from 1883.

Drought terminations show considerable spatio-temporal variability. For example, the 1988-93 event had a notably uneven temporal evolution, with the transition to drought termination occurring early in the drought followed by a long drought termination phase for catchments in South-west UK and Anglian, whereas shorter drought terminations were apparent in the rest of the country. Fewer droughts have occurred in northern and western areas of the UK than in southern and eastern areas, while drought terminations tend to occur over longer time periods in the south. However, it is important to note the wide range of variability in drought termination characteristics exhibited within individual catchments. Two drought termination events are singled out for more detailed analysis: 1995-98, the most widespread event since the 1970s; and 2009-12, reported as unprecedented in the historical record (Parry et al. 2013).

## 4.2   Event analysis: 1995-98

Drought in 1995-98 affected all but one of the study catchments (Fig. 5; left), offering the best opportunity to analyse the spatial variability of drought termination within a single, severe event. The overall duration of drought was up to three years in the south and east in the UK but generally shorter in the north. There were two distinct patterns of drought termination. In the north and west, the drought termination phase began within six months of the start of drought development and long drought termination phases (three or more seasons) followed in 13 catchments. In contrast, drought termination started almost two years later in 25 catchments, mainly in the south and east. The transition to drought termination was generally spatially

coherent across North & Central Wales, Midlands, South-west UK and Southern England, with the exceptions of the Conwy (NCW), Tywi (SWUK) and Great Stour (SE).

Drought termination durations were generally longer (by six to nine months) for catchments in Southern England and Anglian regions (Fig. 5; top right). Conventionally referred to as the 1995-97 drought in the literature (e.g. Marsh et al. 2013; Spraggs et al. 2015), it was the second half of 1998 before catchments in parts of lowland England (e.g. the Warwickshire Avon, Colne, Thames, Itchen and Dorset Avon) had completed the drought termination phase. The drought termination rate displayed a west-east divide in 1995-98, particularly apparent for Wales, southern and eastern England, and the Midlands (Fig. 5; middle right). Whilst much of Wales and south-west England exhibited drought termination rates of 16-32% per month, this decreased to less than 8% per month across large areas of south-eastern England. Further north, the pattern was more mixed. Two-season drought terminations (Fig. 5; bottom right) generally were confined to the far northern parts of Scotland and England. Three-season drought terminations started in the autumn in Scotland and in the winter in Wales, south-western England and the Midlands. Long drought terminations (more than eight months across four or more seasons) in many catchments in Western Scotland, Northern Ireland, North-west England, North-east England, Anglian and Southern England prevented an assessment of drought termination seasonality.

### 4.3  Event analysis: 2009-12

In contrast to the 1995-98 event the 2009-12 drought was regional, primarily affecting North & Central Wales, South-west UK, Anglian, Southern England and the Midlands. The temporal sequencing of drought termination was also more regionally variable than in 1995-98. Drought terminations began much sooner (early summer 2010) in North-west England, and had ended whilst drought continued to develop further south (Fig. 6; left). Droughts terminations started in South-west UK up to a year before those in Anglian and the Midlands. In Anglian, Southern England and the Midlands, drought termination began in winter 2011/12 or spring 2012 and ended in late spring or early summer 2012. The end of the drought termination phase was much more spatially coherent in 2009-12 than in 1995-98.

Drought termination durations in 2009-12 were generally six months or less (Fig. 6; top right), much shorter than those for 1995-98. There was a gradient in drought termination duration from north-east to south-west across the affected catchments. The shortest durations (1-3

months) occurred across southern and eastern England and the Midlands, but lasted longer (10-18 months) for catchments in the south-west of England and Wales. The highest drought termination rates (more than 32% per month) occurred in the largest catchments, whilst lower values (less than 16% per month) were restricted to smaller catchments in Northern Ireland, North-east England and the far south of England (Fig. 6; middle right). Drought termination rates in 2009-12 showed a similar gradient to drought termination duration. There was more uniformity in drought termination rate across the drought-affected area for 2009-12 than in 1995-98, and drought terminations were generally more abrupt in 2009-12.

There was greater seasonality for the 2009-12 drought (Fig. 6; bottom right) than for the 1995-98 event because drought terminations were generally shorter and started at different times. Catchments in southern and eastern England, the Midlands and north Wales experienced drought terminations in spring and/or summer. Drought terminations in the winter months were uncommon for the 2009-12 event. Winter drought terminations were restricted to the Warwickshire Avon (Midlands) and smaller catchments in the Anglian and Southern England regions.

## 4.4   Drought termination and catchment properties

The above analysis offers a qualitative assessment of the impact of catchment type on drought termination characteristics. Longer drought termination durations occurred in groundwater influenced catchments of southern and eastern England (e.g. the Stringside in Anglian and the Itchen and Dorset Avon in Southern England) during both 1995-98 and 2009-12, although this link does not apply for all identified drought termination events in the historical record. However, the synchronicity of the end of drought termination in spring 2012 (Fig. 6; left), when compared to the incoherent end of drought termination in 1995-98 (Fig. 5; left), suggests that catchment properties are less influential during abrupt drought terminations than during gradual events.

Spearman correlations ($r_s$) between drought characteristics (magnitude, termination duration and termination rate) and five catchment properties (catchment area, median elevation, SAAR6190, BFI and urban extent) were calculated from the inventory of events. Correlations were assessed for individual drought events ($n$=459) as well as for catchment averaged values ($n$=52) (Table 1) and were considered statistically significant where $p<0.05$.

The highest $r_s$ (-0.48; $p<0.001$) was found for catchment average drought termination duration and median elevation, suggesting that upland catchments tend to experience shorter drought terminations. A similar correlation value was found between SAAR6190 ($r_s$=-0.40; $p$=0.004) and drought termination duration, most likely due to the association between elevation and rainfall ($r_s$=0.71; $p<0.001$). Drought termination rate and urban extent are negatively correlated ($r_s$=-0.43; $p$=0.002). This association may be influenced by a groundwater signal that is generally stronger in the more urbanised south and east of the UK, although $r_s$ values for BFI and drought termination rate are small (-0.12; $p$=0.412).

Spearman correlations were also derived for a subset of the study catchments, with 17 out of the 52 meeting the criteria of at least 10 identified drought termination events (Table A1). A statistically insignificant correlation was found between catchment average drought termination rate and BFI ($r_s$=-0.36; $p$=0.156). This is consistent with the expectation of faster drought termination rates (i.e. more abrupt drought endings) in lower BFI (i.e. more responsive) catchments. For this subset of catchments, relationships between drought termination duration and both elevation and rainfall again corresponded to the highest values of $r_s$, but the linkages between urban extent and both drought termination duration ($r_s$=0.49; $p$=0.049) and drought termination rate ($r_s$=-0.47; $p$=0.057) were comparable.

For correlations between the properties of the drought development phase and drought termination characteristics, significant relationships were detected for drought development duration with both drought termination duration ($r_s$=-0.30; $p<0.001$) and drought termination rate ($r_s$=0.28; $p<0.001$). This implies that sustained periods of drought development tend to be succeeded by shorter and more abrupt drought terminations. Relationships with catchment average drought development characteristics are not statistically significant, but assessments with the larger individual event dataset found that most associations (e.g. between drought magnitude and drought termination duration, or between drought development duration and drought termination rate) are significant.

## 5  Discussion

This study has systematically discretised drought terminations in historical river flow records for the UK for the first time. The detection method identified 459 drought events across 52 study catchments, providing a comprehensive inventory for further analysis of the historical variability of drought termination. Two aspects are explored here: a preliminary assessment of

linkages between drought termination characteristics and catchment properties, including
features of the preceding drought development phase (informed by the correlation analysis
above); and a re-appraisal of drought termination characteristics in 2009-12 within a broader
hydrological context.  In addition, this section also corroborates the inventory of drought events
and their terminations against existing work in the research literature, and considers the
influence of the data and methodology on the results.
## 5.1   Drought termination characteristics and catchment properties
The spatio-temporal variability in drought termination within individual events (Fig. 4; Fig. 5;
Fig. 6) reflects the amount and timing of rainfall as well as its modulation by local catchment
properties.  This supports the findings of earlier studies that show hydrological drought
termination to be more spatially variable than drought development, owing to the heterogeneity
of catchment characteristics (e.g. Nkemdirim & Weber 1999; Bell et al. 2013; DeChant &
Moradkhani 2015).  However, the balance between the importance of rainfall distribution (in
space and time) and catchment properties varies.  In responsive catchments rainfall receipt will
largely determine drought termination, whilst characteristics of the catchment may have more
influence in those that are less responsive.
Some of the strongest correlations were found between drought termination duration and both
elevation and catchment average rainfall (SAAR6190).  This is likely to be because catchments
in wetter upland areas of the UK are typically impermeable and responsive to rainfall,
translating to shorter drought terminations.  The correlations between urban extent and both
drought termination duration and drought termination rate imply that drought terminations tend
to be longer and more gradual in catchments with larger urban areas.  This contradicts the
expectation that typically impermeable urban areas may exhibit more abrupt drought
terminations.  The more urbanised catchments of the UK are generally in the south-east with
more permeable geology and it may be that lower responsiveness to rainfall negates the impact
of the urban extent.  Note also that the urban extent data are based on satellite imagery from
1998-2000 and, therefore, do not reflect the changing proportion of a catchment as built area
outside of this short period.  Further research could be undertaken to assess the impact of
increasing urbanised area on changes in drought termination characteristics within certain study
catchments under increasing development pressure (e.g. the Great Stour in Southern England).
The BFI is widely regarded as a proxy for groundwater influence in the UK. However, water
storage in lakes and seasonal snow cover can also be locally important, with BFI values of 0.43-
0.60 for the Spey, Deveron, Scottish Dee and Naver in northern Scotland despite negligible
groundwater influence. Whilst these impermeable catchments typically respond rapidly to
rainfall, catchments with similar BFI values in areas of groundwater influence further south are
less responsive. BFI is often considered to reflect catchment responsiveness, but the presence
of lakes and/or snow cover in some responsive catchments of the north and west of the UK
mean that elevation is a better indicator of the spatial variability of geology in the UK than BFI.
This may explain why correlations between drought termination characteristics and elevation
are stronger than those with BFI. By excluding catchments in Scotland that exhibit mismatches
between BFI and responsiveness (through the use of the subset of 17 catchments with at least
ten events), the correlation analysis found a stronger association between drought termination
rate and BFI. This linkage, as well as the qualitative observation of longer drought terminations
in groundwater influenced catchments, is consistent with previous studies that report longer
duration drought termination in soil moisture and groundwater levels (e.g. Eltahir & Yeh 1999;
Thomas et al. 2014).
Stronger relationships identified in the larger dataset between drought development and drought
termination characteristics suggest that catchment averaging of metrics prior to correlation
analysis may smooth out unique associations, resulting in information loss and obscuring some
signals. A weak negative (but statistically significant) correlation was found between drought
magnitude and drought termination duration, contrary to the pattern reported for two multi-year
droughts in the US (Nkemdirim & Weber 1999). The most important linkages were between
drought development duration and both drought termination duration and drought termination
rate.
## 5.2   Validating the chronologies of drought and drought termination
The rarity of national scale droughts over the instrumental period (i.e. 1970s onwards) – limited
to events in the mid-1970s and mid/late 1990s – corroborates previous work on regional drought
in Europe (Hannaford et al. 2011). The locus of the 1988-93 drought in the south-east of the
UK confirms the chronology of Marsh et al. (2007). Time series of regional drought
(Hannaford et al. 2011) identify a number of minor periods of river flow deficiency in the
decade following the 1975-77 event but such episodes were not prolonged or severe enough to
be detected in this study. However, the 1962-64 drought was identifiable here despite the

limited spatial coverage of river flow data. This event has been cited as an important multi-year drought at both UK and European scales (Parry et al. 2012). Similarly, Marsh et al. (2007) identify both the 1959 event and 1890-1910 'Long Drought' when cataloguing major droughts in the UK. Whilst the use of standardised indicators (e.g. Hannaford et al. 2011) identifies the same amount of time under deficit conditions in each region, it is clear that streamflow deficiencies are fewer but more prolonged in southern and eastern areas of the UK, confirming the results presented herein.

Validating the drought termination phases in Figure 4 is less straightforward because of the relative lack of focus in the literature on the end of a drought relative to its other characteristics. Some of the longest drought termination durations correspond to the 1988-93 drought, particularly for the Witham in Anglian region, reflecting previous findings that the recovery from this drought was generally prolonged and particularly so in groundwater influenced catchments (Marsh et al. 1994). Conversely, the abrupt nature of drought terminations corresponding to the 1975-77 event, evident in the chronologies presented herein, has been widely reported in the literature (e.g. Doornkamp et al. 1980; Rodda & Marsh 2011).

## 5.3 Drought termination rate for 2009-12 in a historical context

The rate of drought termination in 2009-12 was particularly abrupt – more so than any other event identified in the post-1970 common data period. Almost a third (nine out of 31) of the drought-affected catchments in 2009-12 registered new maxima for drought termination rate (Table 2). For the Severn, the drought termination in 2009-12 was almost four times more abrupt than any other event since records began in 1929. This ranks amongst the top five most abrupt drought terminations for *any* event in *any* of the 52 study catchments (*n*=459) although lagging substantially behind the most abrupt drought termination in this same dataset: the Whiteadder Water (Eastern Scotland) in 2004-07, which was a third larger than the second ranked event. Drought magnitudes in 2009-12 were not exceptional but it was the differences between drought magnitudes and termination magnitudes over such short drought termination durations that were particularly noteworthy in establishing new maximum drought termination rates. This suggests that exceptional rainfall totals accumulated over short durations (assessed as greater than a 100-year return period; Bell et al. 2013) were more important than the severity of the preceding drought.

Research conducted in the immediate aftermath of the 2009-12 event suggested that the drought termination was unprecedented in the historical record (Parry et al. 2013; Marsh et al. 2013). However, the assessment of the rarity of such abrupt transitions was based on ratios between average river flows over arbitrarily defined periods (May-July and the preceding December-March; Marsh et al. 2007). The more systematic approach adopted here allows an objective re-appraisal of the historical context across all timeframes. Although the drought termination event in 2009-12 remains the most abrupt on record for the Thames (Table 2), there were three other comparably abrupt drought terminations between 1883 and 1930. This suggests that the rarity of the 2009-12 drought termination may have been overstated (in the specific case of the Thames).

The drought termination phases in 2009-12 and 2004-07 were the most abrupt on record for 17% and 15% of the 52 catchments, respectively; no other event registered new maxima in more than 10% of catchments, although this is difficult to assess consistently prior to 1970 due to limitations in data availability. These recent severe multi-year droughts featured consecutive dry winters (Wilby et al. 2015), supporting the view that long droughts result in more abrupt drought termination phases. However, the possibility that drought termination rates are becoming more abrupt warrants further exploration.

The wide variation in drought termination rates both *between* and *within* catchments suggests that different drought termination mechanisms are at work. Drought termination reflects a complex interplay of the specific hydroclimatic conditions with local catchment properties, even for groundwater influenced permeable catchments (in which the rainfall signal is substantially modulated by geology). Groundwater drought termination has been observed to be much slower than drought development in the western US (Bravar & Kavvas 1991). Whether this applies to individual events in groundwater influenced catchments in this study would depend on the extent to which deficits have propagated to groundwater. The artificial depletion of groundwater aquifers in Southern England may also have impacted drought termination characteristics in some catchments (e.g. the Itchen). The approach adopted in this study could be extended to groundwater level records as a further line of research. Similar variability in drought terminations was reported by Bonsal et al. (2011), and was attributed by Kam et al. (2013) to differences in rainfall intensity determined by the synoptic conditions (e.g. tropical cyclones).

## 5.4  Drought termination seasonality for 2009-12 in a historical context

The drought termination in 2009-12 occurred through the spring and early summer, an unusual but not unprecedented event. Only nine of the 459 drought terminations occurred entirely in spring or in summer. Five of these nine relate to the 2009-12 event (the Severn, Trent, Derwent and Witham in spring, and the Colne in summer). With the exception of the Severn, the drought termination in 2009-12 is the only single season event in the historical record for each catchment. Drought terminations across both spring and summer are similarly rare. Of the 13 events (out of 459) with spring-summer drought termination seasonality, five occurred in 2009-12 (the Yscir, Exe, Thames, Itchen and Sydling Water; Fig. 6, bottom right). Of the remaining eight events, no other drought termination is represented by more than two catchments. For the Thames, the only previous example of a drought termination entirely within the spring and summer was in 1888. Other studies have also found that it is unlikely that multi-season droughts will terminate in two seasons or less (Karl et al. 1987).

Rather than simply the wettest season, it is the season with the greatest potential for large positive rainfall anomalies that is most likely to facilitate drought termination (Karl et al. 1987; Mo 2011). In the UK these two factors conincide, hence, winter provides the greatest likelihood for drought termination (Van Loon et al. 2014). The larger evaporative demand in summer reduces the effectiveness of all but the most extreme rainfall, explaining the tendency for drought terminations in the winter half-year. Of the 459 drought terminations, single season events were more common in autumn (eight) and winter (eight) than in spring (six) and particularly summer (three).

At regional scales, variation in drought termination seasonality is likely to be determined by catchment properties, such as storage causing lagged responses. For catchments in Scotland, the influence of snow may also influence drought termination. Where seasonal snowpacks exist, winter drought terminations may be delayed until the snowmelt season (Van Loon et al. 2014). However, the large variability of drought termination characteristics and the moderate to weak correlations with catchment properties imply that a range of physical processes are involved. At national or continental scales, larger scale drivers such as El Niño and La Niña events in the Pacific (e.g. Tomasella et al. 2011; Marengo & Espinoza 2015), switches in Atlantic temperatures (Wilby 2001; Folland et al. 2015) and tropical cyclones (e.g. Kam et al. 2013; Patterson et al. 2013) have been shown to be a factor in drought termination events. Further research is required to assess the extent to which changes in these and other synoptic

drivers might be influencing the seasonality of drought terminations in the UK. For instance,
Matthews et al. (2015) report relatively low frequencies of summer cyclones in the period 1961-
90 but a marked resurgence in counts since the 1990s.
## 5.5 Impact of methodology and data on results
Although the detection procedure utilised herein applied consistent rules, the parameter values
used to define a drought and its phases can influence the resulting chronology. This is
illustrated by the sensitivity analysis (Fig. 3) and has been reported by other studies (e.g.
Patterson et al. 2013). Drought termination phases following shorter drought developments,
for example driven by summer heatwaves, would not be well represented by the parameter
settings used in this study. This is because the parameters which determine the initiation of
drought development ($D$ and $R$) require below average river flows for at least nine of ten
consecutive months, a timeframe which is too prolonged to adequately characterise typical
single season drought events. In addition, events in the more hydrologically responsive north
and west of the UK might be less well represented because droughts in these wetter regions are
typically shorter than multi-season in duration. However, the spatial variability in the number
of identified droughts is consistent with the levels of service set by regional water companies,
with drought-induced water restrictions expected more frequently in the south-east of the UK
than in the north. Nevertheless, there is a need to more comprehensively assess the sensitivity
of derived chronologies of drought termination to the choice of detection parameters.
The monthly time step used in this study may also be limiting. Drought termination can occur
rapidly, perhaps within a few days in some instances of intense cyclonic activity. Under these
circumstances, monthly data may obscure accurate definitions of the end of drought termination
or underestimate the drought termination rate. In addition, the use of a monthly average flow
threshold is higher than those usually applied in threshold-based studies. Low flow thresholds
such as $Q_{70}$ (Hisdal et al. 2001) and $Q_{80}$ (e.g. Tallaksen et al. 2009) have been widely used in
the literature, and threshold levels between $Q_{70}$ and $Q_{90}$ are generally considered appropriate
(Fleig et al. 2006). The use of an average flow threshold would be expected to increase the
overall duration of drought (as illustrated by Tallaksen et al. 1997) as well as the drought
development and drought termination phases. However, applying a lower threshold would sub-
divide well-known multi-year drought events (e.g. 1995-98 and 2009-12 from this study) into
a number of more severe episodes each with their own drought termination. In order to focus
on multi-season to multi-year droughts, a higher threshold is required. A previous study that

applied thresholds between $Q_{50}$ and $Q_{90}$ found that a higher threshold level identified more multi-year droughts (Tallaksen et al. 1997). It is acknowledged that the suitability of different thresholds is specific to individual perceptions or applications.

The approach utilised in this study focuses on the status of river flows, which can increase substantially over relatively short timescales and replenish water supplies rapidly without having to account for a deficit that has accumulated during the drought development phase. However, it is acknowledged that deficit volume approaches (in which the accumulated volume of water 'lost' during drought development is recovered) may be important for studies which focus on the overall water balance.

The potential influence of abstractions from surface and groundwater sources during drought development may artificially extend the duration of the drought termination phase. The catchments used in this study include some of the largest in the UK in order to maximise spatial coverage, and few of these could be described as near-natural. Abstractions to meet higher water demand during drought development, particularly during heatwave conditions, combine with lower natural recharge. Drought-terminating rainfall must account for this 'anthropogenic deficit' in addition to the natural hydrological deficit. There is a regional bias in the anthropogenic influence on river flows, with more impacted catchments in the south and east of the UK and more near-natural catchments in the north and west. Whilst this spatial pattern also reflects the number of droughts identified, the selection of parameters that favour major multi-season droughts is probably more influential. The use of monthly mean river flows may also dilute the impact of artificial influences on individual days.

## 6 Conclusions

For the first time, terminations of multi-season to multi-year droughts in the UK have been systematically identified and characterised. This study detected 459 events in 52 catchments covering a range of geographical settings, and provides chronologies of both drought development and drought termination phases. This information provides a new perspective on the historical variability of drought termination in the UK that is potentially useful for water resource managers and researchers in a range of fields including ecology, geomorphology and water quality. It is hoped that characterising 459 drought termination events will underpin further research into any emerging trends and provide the basis for the development of a drought termination typology. It should be noted that the chronology of drought termination presented

herein has been derived using parameters that were informed by a sensitivity analysis and ensuring a focus on multi-season to multi-year droughts in the UK. For other applications across a range of locations and/or considering alternative definitions of droughts, it is recognised that alternative parameters may be required.

Investigations into the link between drought termination characteristics and catchment properties or drought development characteristics would be strengthened by a larger sample of events. Stronger correlations were found for catchment average drought termination metrics when using the subset of catchments with at least ten identified events, although this subset is biased towards catchments with longer records predominantly in southern and eastern areas of the UK. The BFI is not an adequate predictor of the responsiveness of a catchment. Further exploration of potential linkages between drought termination characteristics and catchment properties should seek to use variables which are more closely related to river flow responsiveness than BFI (e.g. a flashiness index; Baker et al. 2007). The use of potential associations between drought termination characteristics and those of the preceding drought development phase by water resource managers is constrained by weak to moderate correlations and requires further research before useful conclusions can be drawn. Ideally, coupled land-atmosphere model experiments would be performed to explore possible links between drought duration or magnitude and terminating rainfall mechanisms.

The identification and characterisation of 459 drought terminations has provided a comprehensive historical context within which to place the notable 2009-12 event. This illustrates the variability of drought termination characteristics in the UK, re-assessing the conclusion (based on a subset of newsworthy examples) that droughts tend to terminate abruptly. The long-term context could be improved further through the use of river flow reconstructions (e.g. Jones and Lister 1998; Jones et al. 2006) to 'fill in the grey space' in Fig. 4, which represents the best historical perspective provided by available observed data. The method used in this study has the flexibility to produce similarly comprehensive chronologies of drought termination in groundwater level records, water quality metrics or ecological indices, to trace the propagation of drought termination throughout the river system and hydrological cycle. Drought termination in river flows and groundwater levels may not synchronise even within the same catchment due to lagged response times. Hence, even when a drought terminates abruptly with severe river flooding, (contrary to public expectations) water

restrictions may not be removed until groundwater levels respond. The complexities associated
with this propagation of drought termination require further research.
**Appendix A**
Table A1. Metadata for the 52 study catchments. The subset of 17 catchments referred to in
sections 4.4 and 5.1 is indicated with asterisks (*).

| Region | Catchment | Record length (years) | Area (km$^2$) | Median elevation (m) | SAAR6190 (mm) | BFI | Urban extent (%) |
|---|---|---|---|---|---|---|---|
| W Scotland | Naver | 37 | 477 | 187 | 1384 | 0.43 | 0.0 |
| W Scotland | Carron | 35 | 138 | 342 | 2620 | 0.26 | 0.0 |
| W Scotland | Nevis | 32 | 69 | 518 | 2912 | 0.27 | 0.1 |
| W Scotland | Clyde | 51 | 1903 | 252 | 1129 | 0.46 | 3.0 |
| W Scotland | Ayr | 38 | 574 | 212 | 1214 | 0.30 | 0.6 |
| W Scotland | Cree | 51 | 368 | 212 | 1760 | 0.28 | 0.2 |
| W Scotland | Nith | 37 | 477 | 288 | 1460 | 0.39 | 0.2 |
| E Scotland | Findhorn | 56 | 782 | 408 | 1064 | 0.40 | 0.0 |
| E Scotland | Spey* | 62 | 2861 | 420 | 1120 | 0.60 | 0.1 |
| E Scotland | Deveron* | 54 | 955 | 209 | 928 | 0.57 | 0.2 |
| E Scotland | Scottish Dee* | 85 | 1370 | 508 | 1109 | 0.53 | 0.1 |
| E Scotland | Tay | 62 | 4587 | 395 | 1425 | 0.65 | 0.2 |
| E Scotland | Forth | 33 | 1036 | 180 | 1752 | 0.41 | 0.0 |
| E Scotland | Whiteadder Water | 45 | 503 | 230 | 813 | 0.51 | 0.2 |
| E Scotland | Tweed | 52 | 4390 | 255 | 955 | 0.52 | 0.3 |
| N Ireland | Mourne | 32 | 1844 | 153 | 1288 | 0.39 | 0.3 |

| | | | | | | | |
|---|---|---|---|---|---|---|---|
| N Ireland | Faughan | 38 | 273 | 173 | 1219 | 0.47 | 0.4 |
| N Ireland | Lagan | 42 | 492 | 95 | 916 | 0.43 | 3.2 |
| NW England | Eden | 47 | 2287 | 210 | 1183 | 0.49 | 0.8 |
| NW England | Kent | 46 | 209 | 205 | 1732 | 0.41 | 1.8 |
| NW England | Ribble | 54 | 1145 | 198 | 1353 | 0.34 | 3.7 |
| NE England | South Tyne | 52 | 751 | 333 | 1148 | 0.34 | 0.2 |
| NE England | Tees | 58 | 818 | 370 | 1141 | 0.34 | 0.4 |
| NE England | Ure | 56 | 915 | 264 | 1118 | 0.39 | 0.8 |
| NE England | Derwent | 41 | 1586 | 102 | 765 | 0.70 | 0.8 |
| N&C Wales | Conwy | 50 | 345 | 328 | 2055 | 0.28 | 0.1 |
| N&C Wales | Welsh Dee | 77 | 1013 | 347 | 1369 | 0.54 | 0.4 |
| N&C Wales | Severn* | 93 | 4325 | 127 | 913 | 0.53 | 2.0 |
| N&C Wales | Teme | 44 | 1480 | 191 | 818 | 0.55 | 0.7 |
| N&C Wales | Wye* | 78 | 4010 | 199 | 1011 | 0.54 | 0.7 |
| Midlands | Trent* | 56 | 7486 | 118 | 761 | 0.64 | 10.5 |
| Midlands | Warwickshire Avon* | 78 | 2210 | 96 | 654 | 0.51 | 4.9 |
| SW UK | Tywi | 56 | 1090 | 220 | 1534 | 0.47 | 0.2 |
| SW UK | Yscir | 42 | 63 | 361 | 1299 | 0.46 | 0.0 |
| SW UK | Tone | 53 | 202 | 120 | 966 | 0.60 | 1.6 |
| SW UK | Torridge* | 54 | 663 | 146 | 1186 | 0.38 | 0.4 |
| SW UK | Exe* | 58 | 601 | 235 | 1248 | 0.50 | 0.6 |
| SW UK | Dart | 56 | 248 | 347 | 1765 | 0.52 | 0.7 |

| | | | | | | | |
|---|---|---|---|---|---|---|---|
| SW UK | Warleggan | 45 | 25 | 232 | 1442 | 0.70 | 0.2 |
| SW UK | Sydling Water* | 45 | 12 | 190 | 1032 | 0.88 | 0.5 |
| Anglian | Lud | 46 | 55 | 89 | 699 | 0.90 | 2.2 |
| Anglian | Witham* | 55 | 298 | 91 | 614 | 0.69 | 3.5 |
| Anglian | Bedford Ouse* | 81 | 1460 | 101 | 636 | 0.53 | 3.5 |
| Anglian | Stringside | 49 | 99 | 20 | 629 | 0.84 | 0.7 |
| Anglian | Wensum | 45 | 398 | 57 | 684 | 0.75 | 1.3 |
| Anglian | Colne* | 55 | 238 | 68 | 566 | 0.52 | 2.2 |
| S England | Thames* | 131 | 9948 | 100 | 706 | 0.63 | 6.6 |
| S England | Great Stour* | 50 | 345 | 75 | 747 | 0.70 | 3.2 |
| S England | Bull | 36 | 41 | 58 | 820 | 0.37 | 0.9 |
| S England | Itchen | 56 | 360 | 107 | 833 | 0.96 | 2.9 |
| S England | Dorset Avon* | 49 | 324 | 129 | 745 | 0.91 | 1.3 |
| S England | Stour* | 41 | 1073 | 83 | 861 | 0.64 | 2.0 |

## Author contribution

S. Parry devised the approach, selected the catchments and coordinated the writing of the paper. R. L. Wilby provided input on the structure and content of the paper and the impetus for the correlation analysis. C. Prudhomme and P. J. Wood provided feedback on the different paper structures and content. All authors contributed to the manuscript writing and commented on the analyses.

## Acknowledgements

This research was funded through the Learning & Development programme at the Centre for Ecology & Hydrology (CEH), as well as the Natural Environment Research Council's (NERC)

'Analysis of historic drought and water scarcity in the UK' (NERC Grant Ref.: NE/L01016X/1)
and 'Improving predictions of drought to inform user decisions (IMPETUS)' (NERC Grant
Ref.: NE/L010267/1) projects.  River flow data and catchment metadata were provided by the
UK National River Flow Archive at CEH.  The manuscript was improved following valuable
feedback provided by two reviewers, Henny van Lanen and an anonymous reviewer.  The
authors would also like to thank Katie Muchan, Filip Kral, Lucy Barker and Shaun Harrigan
(all CEH) for their assistance with river flow data and metadata, spatial data, graphics, and
statistics, respectively.

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

Table 1. Spearman correlations between drought termination characteristics and both catchment
properties and drought development characteristics. Correlations are presented for individual events
(rows $n$=459) and for catchment mean drought characteristics (rows $n$=52). Asterisks (*) denote
statistical significance at the 95% confidence level. Drought termination characteristics are defined as:
DTD = drought termination duration; DTR = drought termination rate. Drought development
characteristics are defined as: DDD = drought development duration; DM = drought magnitude.
Catchment properties are denoted as: SAAR6190 = Standard-period Average Annual Rainfall for 1961-
90; BFI = Base Flow Index.

| | | Catchment properties | | | | | Drought development characteristics | |
|---|---|---|---|---|---|---|---|---|
| | $n$ | Area | Median elevation | SAAR 6190 | BFI | Urban extent | DDD | DM |
| DTD | 459 | -0.03 | -0.15* | -0.12* | 0.04 | 0.14* | -0.30* | -0.19* |
| DTD | 52 | -0.23 | -0.48* | -0.40* | 0.13 | 0.40* | 0.03 | -0.06 |
| DTR | 459 | 0.02 | 0.12* | 0.12* | -0.18* | -0.15* | 0.28* | -0.04 |
| DTR | 52 | 0.11 | 0.22 | 0.12 | -0.12 | -0.43* | 0.01 | -0.19 |

Table 2. Catchments for which the drought termination rate during the 2009-12 event was the largest
of any previous event in the historical record.

| Catchment | Number of drought events | Drought termination rate (% per month) | | Year of drought termination ranking 2nd by drought termination rate |
|---|---|---|---|---|
| | | 2009-12 | Rank 2 | |
| Severn | 16 | 90.6 | 26.5 | 1997 |
| Derwent | 7 | 62.3 | 42.6 | 1976 |
| Trent | 11 | 56.3 | 28.0 | 1959/60 |
| Warwickshire Avon | 20 | 49.6 | 33.7 | 1963 |
| Thames | 35 | 38.1 | 37.2 | 1929/30 |
| Teme | 8 | 33.6 | 29.6 | 1975/76 |
| Sydling Water | 10 | 30.8 | 25.5 | 1974 |
| Itchen | 9 | 21.1 | 12.5 | 1963 |
| Carron | 3 | 18.2 | 11.9 | 2001 |

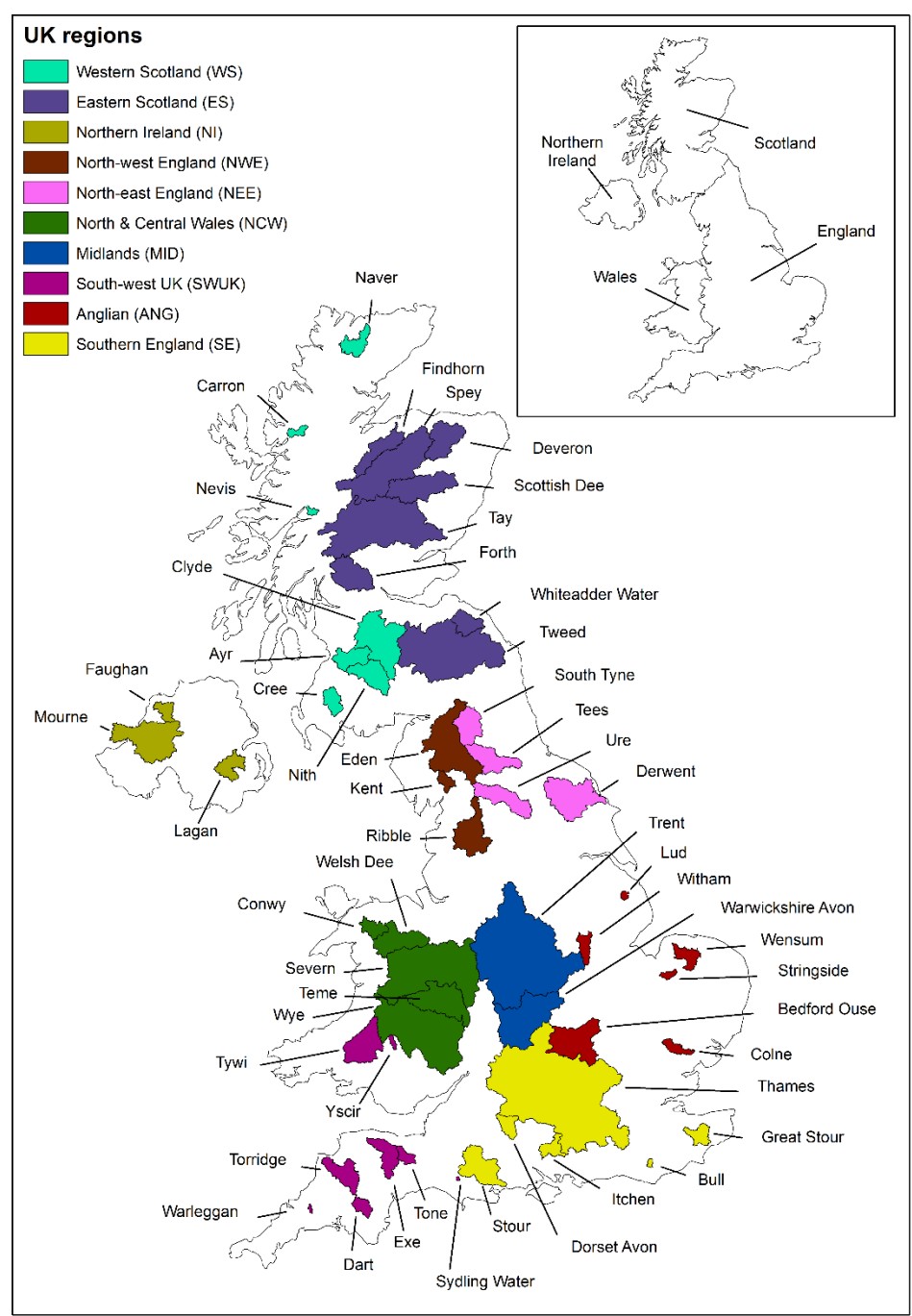

Figure 1. Locations of the 52 study catchments, colour-coded by region. The regions are abbreviated in Fig. 4, Fig. 5 and Fig. 6 as follows: Western Scotland = WS; Eastern Scotland = ES; Northern Ireland = NI; North-west England = NWE; North-east England = NEE; North & Central Wales = NCW; Midlands = MID; South-west UK = SWUK; Anglian = ANG; Southern England = SE. Inset: the constituent countries of the UK.

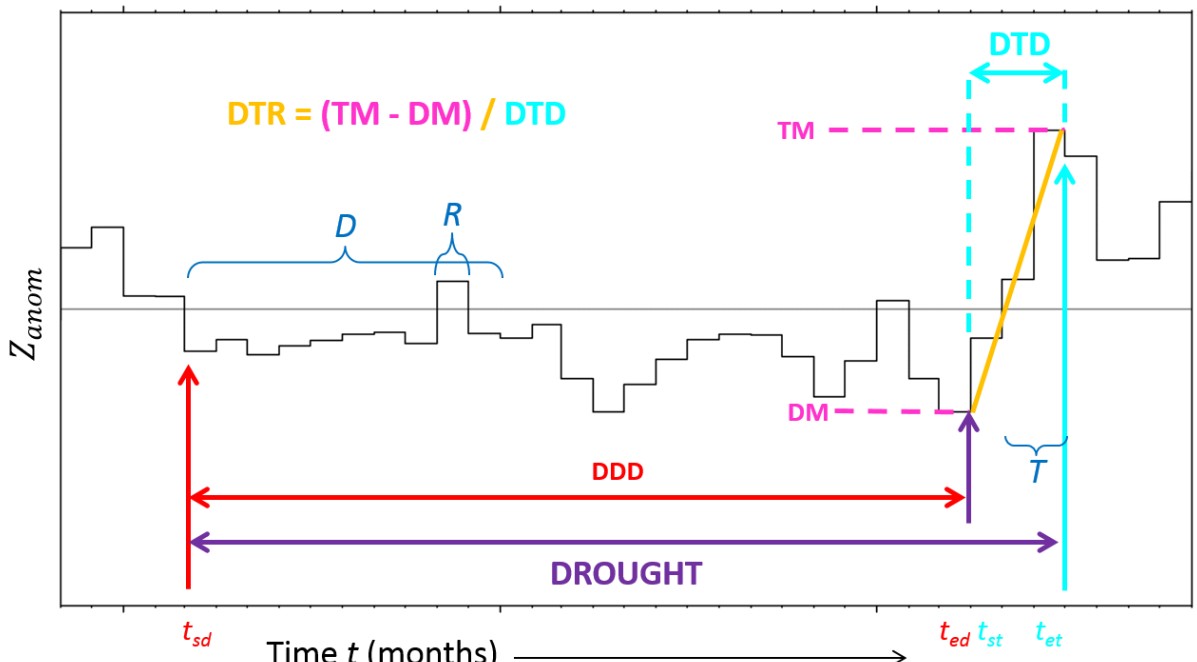

Figure 2. A conceptualisation of drought termination definition and metrics. The three
parameters are as follows: $D$ is the number of months of below average flows required for the
drought development phase to begin; $R$ is the number of months of intermittent above average
flows permitted within $D$; and $T$ is the number of consecutive months of above average flows
required for the end of the drought termination phase. $t_{sd}$ is the time of start of drought
development, $t_{ed}$ is the time of end of drought development, $t_{st}$ is the time of start of drought
termination, and $t_{et}$ is the time of end of drought termination. The grey horizontal line represents
an anomaly of zero, below which flows are below average and above which flows are above
average.

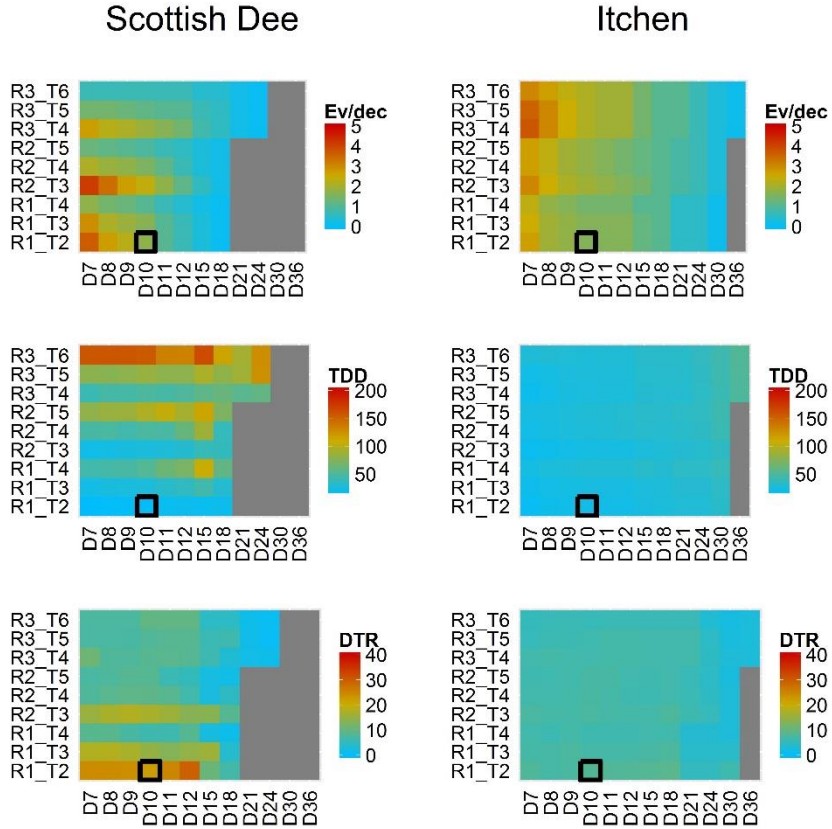

Figure 3. Demonstrations of the sensitivity of drought termination metrics to parameter
selection for the Scottish Dee and Itchen catchments. *D*, *R* and *T* are the three parameters of
the methodology: *D*7-*D*36 are 7- to 36-month durations over which $Z_{anom}$ is negative; *R*1-*R*3
are the number of months (1, 2 or 3) within the *D*-month duration for which $Z_{anom}$ is permitted
to be positive; *T*2-*T*6 are the number of consecutive months (2-6) for which $Z_{anom}$ is positive.
The metrics are: 'Ev/dec' = number of events per decade; TDD = total drought duration
(drought development duration and drought termination duration taken together); DTR =
drought termination rate. The bold box on each response surface shows the combination of
parameters used to derive the drought termination chronologies in this study.

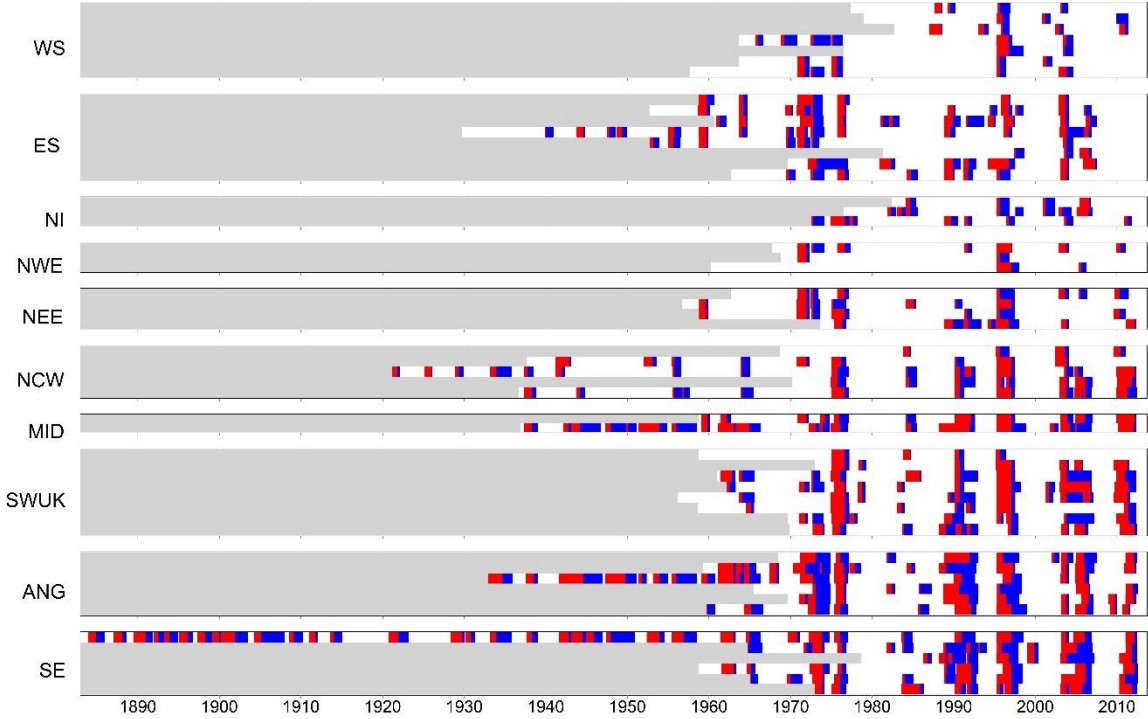

Figure 4. A chronology of drought termination for all 52 study catchments. Red bars indicate
drought development, blue bars indicate drought termination, white bars indicate no drought
development or drought termination, and grey bars signify periods before gauged river flow
records began. On the *x*-axis, a decade (e.g. 1990-2000) is comprised of 120 monthly time steps
and there are 1569 monthly time steps along the entire *x*-axis (January 1883 to September 2013,
inclusive).

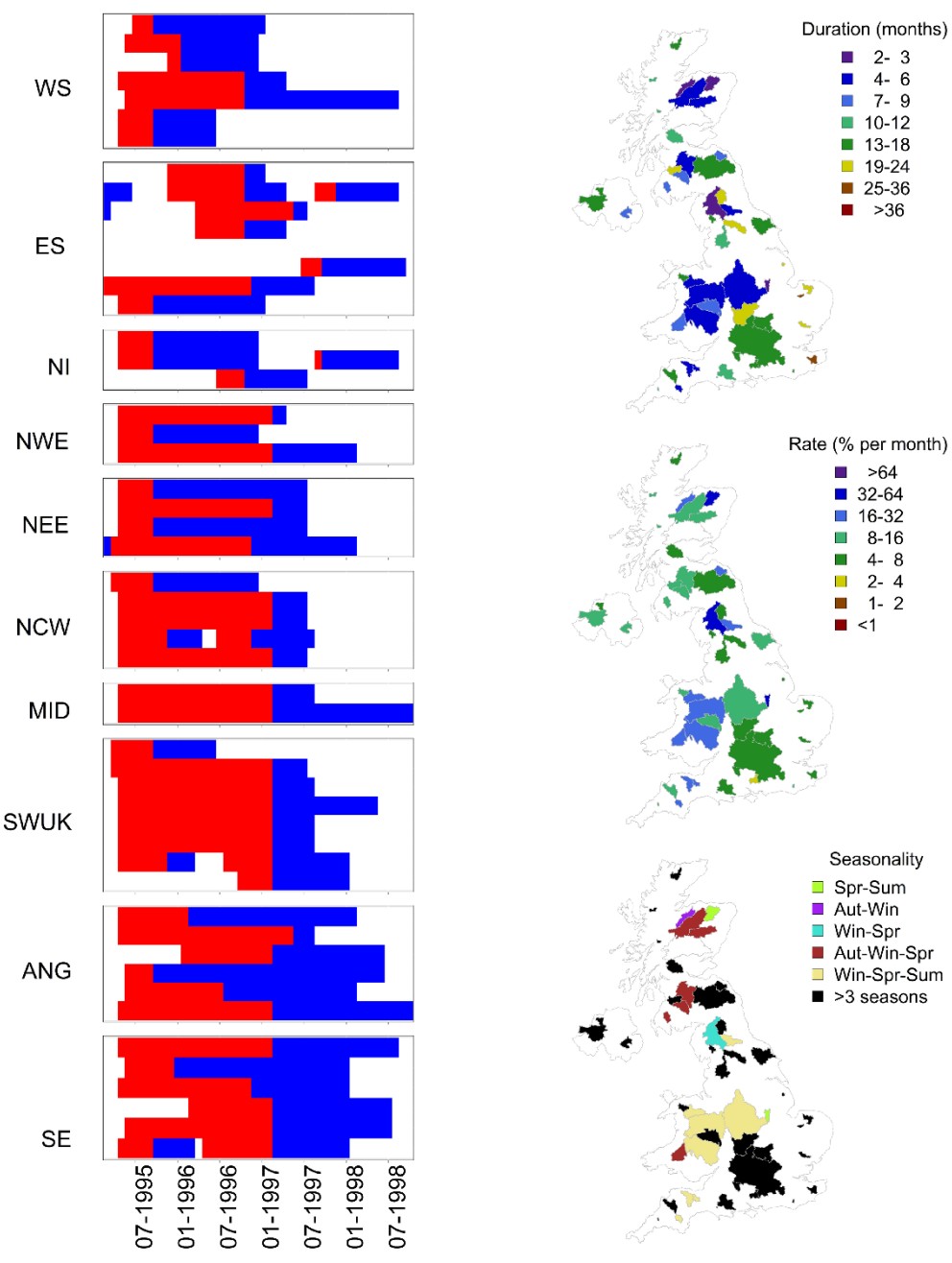

Figure 5. The 1995-98 drought termination: Chronologies of drought development and drought
termination (left); Drought termination duration (top right); Drought termination rate (middle
right); Drought termination seasonality (bottom right).

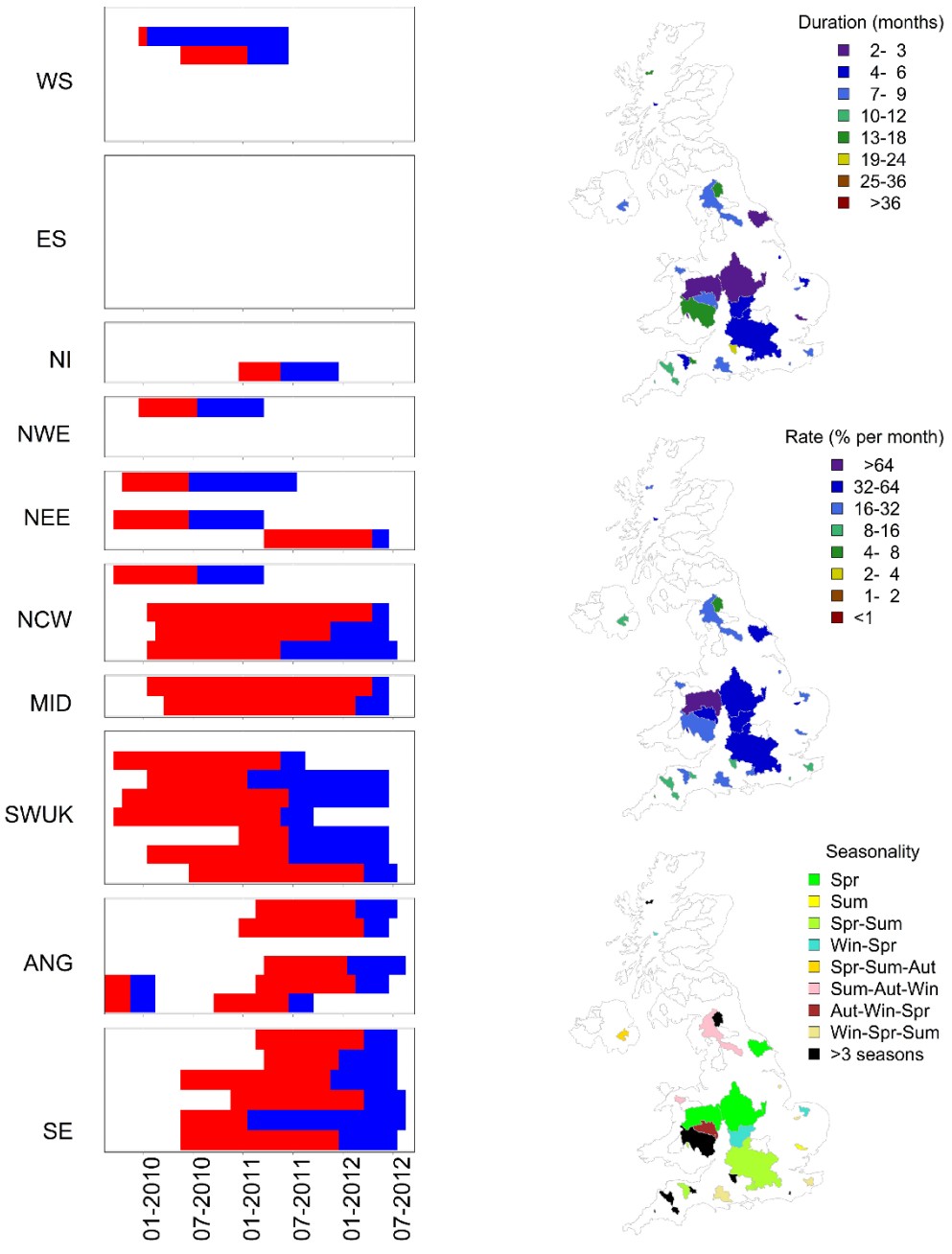

3  Figure 6. The 2009-12 drought termination: Chronologies of drought development and drought

4  termination (left); Drought termination duration (top right); Drought termination rate (middle

5  right); Drought termination seasonality (bottom right).