# Peer review of "A Systematic Assessment of Drought Termination in the"

_Hydrology and Earth System Sciences, 2015_

## Referee Comment (RC1) · Anonymous Referee #1 · 16 Feb 2016

General Comments

This manuscript provides insight into a systematic assessment of drought termination for the United Kingdom. The paper is well written and structured and I particularly like the threshold analysis of drought termination. However, I think the authors over generalize their results and are not clear how this article is truly different from other drought termination research. I think after a moderate revision this article should be published in HESS.

Specific Comments

Page 4 Lines 3–10: The authors argue that their manuscript differs from other drought termination studies by not examining drought termination as "an instantaneous point of time". However, the other studies the authors list do not examine the end of drought as

an instantaneous point in time as they use drought indices that have lags incorporated in the calculations and some of these studies were conducted at the monthly timescale, which is the same timescale used in the this study. I think the authors should revisit this section and be more specific how their study differs from the previous work, which I think it does by using a threshold level but to some extent any drought index (e.g., PDSI) gives thresholds of drought and leaves the defining of the end and beginning of drought up to the user.

Page6, Lines 18–23: I am not convinced that the parameter values used in this study are the most ideal, in particular the R and T variables. Did the authors do any sort of sensitivity analysis to see how this would impact their results? It seems like an R of 2 instead of 1 would dramatically impact the results. Similarly, what if drought was present then the monthly flow went positive for 2 months or even 3 (i.e., T) but then drought returned the next month. I think there needs to be more thought and description into defining R and T and how that influences the results. Page 6 Line 27: Why did the authors not examine if the data was normally distributed? This seems like a weak reason for determine which analysis to conduct. Page 4 Line 23: Why 52 catchments? Need more detail in what went into selecting these catchments.

Page 10 section 4.4: The authors argue that "longer drought termination duration occurred in groundwater influenced catchments of southern and eastern England". While Figure 3 shows this is true for the 1995–1998 and 2009–2012 events however during the 1970s events it would appear this is not true. I'm not sure that there the two most recent events are enough of a sample to draw firm conclusions on the influence of catchment properties on drought termination and think the authors have over generalized their findings. Similarly, on Page 12 Lines 11–16: I think the authors are over generalizing their results. Wouldn't drought termination vary within individual events due to the sporadic nature of rainfall? How do the authors have confidence it is catchment related? The correlations provided in Table 1 show similar relationships between drought termination (DTD and DTR) with SAAR6190 (ie, rainfall) and elevation. Is it

wetter at higher elevations and thus could it just be rainfall variability rather than catchment properties?

Technical Corrections

Abstract, Line 1: The phrase "drought to storms and flooding" is awkward. I recommend removing "storms".

Page 15 Lines 3: I think percentages would be better than "nine and eight of the 52 catchments"
* * *

---

## Referee Comment (RC2) · Dr. van Lanen (Referee) · 17 Feb 2016

**A systematic assessment of drought termination in the United Kingdom**

S. Parry, R.L. Wilby, C. Prudhomme and P.J. Wood

*Hydrol. Earth Syst. Sci. Discuss., doi:10.5194/hess-2015-476, 2016*

In the study, a concept to quantify and map drought termination, including a number of drought termination metrices, was developed. This concept was applied to 52 catchments across the UK representing different physio-geographic conditions (e.g. rainfall, elevation, catchment storage proxy, urbanisation level) to obtain a number of drought termination metrices (duration, termination rate, onset and end seasons). Drought development and termination chronologies, incl. the spatial distribution across the UK, were presented and discussed. Chronologies of two contrasting multi-year droughts were selected (1995-97 and 2009-12) and drought termination metrices were mapped. Results were put in a horizontal context. Eventually, drought termination characteristics from all 459 events and catchments averages were correlated with catchment characteristics and with drought development characteristics. Results were discussed, research gaps were identified and potential links between drought characteristics and catchment characteristics for practical use were provided.

Studies focussing on drought recovery or termination are still rare. The authors managed to improve knowledge on how to conceptualize drought termination, to develop a dataset with associated drought development and drought termination characteristics, and to comprehensively describe the spatial and temporal distribution across the UK. The authors also started to explore underlying mechanism of drought termination through a statistical analysis and draw the attention to abrupt drought termination. The manuscript is well written and the figures (except Fig. 6) and tables adequately support the text. The study is original and it certainly is worthwhile to be published in HESS, but clarifications and improvements are needed before it can be accepted.

Major comments:
1. The authors are one of the first that conceptualize drought termination. Figure 2 clearly illustrates the procedure and suggests that a comprehensive script has been compiled to identify drought development and drought termination characteristics. It enables application of consistent rules. However, as said by the authors, parameter values have to be set to define drought development and termination, and hence determine outcome. Key parameters are the drought initiation parameters $D$ (number of consecutive months that flow has to be below monthly average flow) and $R$ (months within the $D$-month duration with above monthly average flow, to account for minor wet phases during drought development). The third key parameter is $T$ (number of consecutive months with flow above monthly average flow to end a drought). In this study, these parameters are set on 10, 1 and 2 months, respectively, to maximise detection of multi-season events. Another important assumption is that the drought development phase ends at the end of the month with the maximum negative difference between the observed flow and the monthly average flow (this is called the drought magnitude $DM$ in the study). A sensitivity analysis is needed to determine how robust the findings in this study are dependent on the choice of the parameters. The sensitivity analysis can be limited to multi-season droughts (rather high $D$), because characteristics of short-lived are beyond the scope of this study. I am interested in the sensitivity of $R$ and particularly $T$. Consequences of the choice of $DM$ also have to be elaborated. The revised manuscript needs a separate section on the sensitivity analysis in the Discussion.
2. It is assumed that droughts develop and terminate relative to the average monthly flow ($Z_{LTA_m}$). Peters et al. (Eq. 3, Hydrol. Process, 2003) also used average flow, but offered the option (drought criterion $c$) to define a threshold related to average flow. In most drought studies using the

threshold approach, a more extreme flow is selected than in the manuscript, i.e. very often the $Q_{80}$ (flow that is equalled or exceeded in 80% of the time) is used as threshold. I wonder if drought durations (both in the development phase and the termination phase) are not overestimated in this study due to using average monthly flow as a threshold.

3. Authors refer to the importance of drought termination for water management. They propose a duration-based procedure. Duration both of drought development and drought termination certainly are important for water managers, but I wonder if they are not more interested how long it will take to replenish the missing water, e.g. for reservoir of aquifer management. This would imply a deficit volume-based procedure, i.e. not using a pre-defined $T$. I suggest to comment on these different approaches in the Discussion section.

4. In the paper there is emphasis on abrupt drought terminations, like the 2009-2012 event. This is an interesting phenomenon, but I believe for drought management the gradual terminations are more relevant (the long time it takes to let hydrological drought recover). In the paper there is reference to abrupt drought terminations derived from the long Thames flow time series. In the period 1883-2013, four of abrupt terminations (similar to 2009-2012 event) occur, but that is about 10% of all identified multi-season droughts.

5. In the Discussion section drought termination in river flow is compared with terminations in other drought types. For instance, Mo (2001) and Dettinger (2013) use drought in precipitation (SPI6) and soil moisture (PDSI, anomalies). We know that there are differences in drought types due to drought propagation, which does not allow a straightforward intercomparison of terminations of different drought types.

6. I support the authors' view that drought termination is a complex interplay of specific hydroclimatic conditions, catchment properties and human influences (abstractions, urbanisation). As said before, I appreciate very much that they share what we know, but also what we still do not know on underlying mechanism (identification of research gaps). Strength of relationships between drought termination characteristics and catchments characteristics that were found, is weak to moderately weak (Spearman's correlation coefficients are lower than 0.5), although some relationships are significant. It would therefore be wise if authors were to show restraint in exploiting these relationships for practical use. Hence, I would delete paragraphs, like the one about possible critical time thresholds within a drought, which may have important implications for the management of water resources (pg. 13, lines 27-29, pg. 14, lines 1-4).

7. The role of groundwater in drought development and termination has been addressed at several places in the manuscript. I would expect long drought terminations in groundwater-dominated catchments. However, this could not convincingly be proven, even not by a subset of catchments. It appeared that the BFI is not a good proxy for groundwater responsiveness. Some of the catchments have long groundwater time series. It is a pity that authors say that a similar analysis for groundwater is beyond the scope of this study. They could have selected some groundwater hydrographs from a few selected catchments to progress on the role of groundwater (could be included in Section 5.1). This does not imply a comprehensive study of chronologies of drought termination in groundwater level records, as has been done for the river flow. Some droughts in river flow might have been incorrectly terminated, because the drought in groundwater still continued. Water following quick flow paths in the catchment could lead to a temporary flow increase, while it will drop again after some time due to lower than normal groundwater inflow in the river. The sensitivity of $T$ (point 1) might help to increase understanding.

8. In the Conclusions a number of items are addressed (e.g. influence of parameter selection, monthly time step, abstractions) that should be in treated in the Discussion. Only the main findings about the influence should be described in the Conclusions.

9. This study is one of the first on ending of a drought. It would be good to set the terminology right from the beginning. The authors use the term "termination" for the last phase of a drought (decrease of the deficit to zero), whereas I thought that termination is more associated with an instantaneous point in time. I would the use termination for the latter and "recovery" for the last phase.

Minor comments:
- Pg. 2, lines 1-2: I think this is not the right start of the abstract by referring to dramatic / abrupt terminations (see point 4, major comments);
- Pg. 2, lines 21-24: too speculative, because of weak relationships (see point 6, major comments);
- Pg. 3, lines 1-3: long sentence;
- Pg. 3, lines 1-14: I think this is not the right start of the Introduction by referring to violent weather / abrupt terminations (see point 4, major comments);
- Pg. 3, lines 20-26: I think that also reference has be made to the drought typology studies by Van Loon et al. (HESS, 2012; HESS, 2015), in which ending of the drought is also essential;
- Pg. 4, line 6: "…a period of drought termination." sounds a bit strange (see point 9, major comments);
- Pg. 4, line 20: "…40% of the gauged area)" is not an evidence of representative coverage. Rephrase;
- Pg. 5, lines 26-27 and pg. 6, lines 1-3: flow data from the north / west and south /east seemed to have some bias, near-natural and anthropogenic, respectively. You need to come back to this in Discussion;
- Pg. 5, line 22: you cannot say that the LTA refers to 1971-2000, because some flow gauging started in some in 1982. 13 gauging stations have a record that does not include 1970 (Table A.1);
- Pg. 6, line 6: strange to call $RT$ a threshold. Readers will think that a threshold has a specific meaning. Below of above something will happen (e.g. impact). It is usually predefined and derived from other information (e.g. data, unwanted impacts). In the proposed procedure it is the flow ($Z_{anom_t}$) in the second consecutive month above the average monthly flow ($Z_{LTA_m}$). It is just a number that is dictated by the time series, rather than it has generic meaning. I suggest not to use the term threshold in this context;
- Pg. 6, line 12: What is the physical / practical meaning of the drought termination rate (DTR)? A low DTR reflects a long termination duration DTD and/or a large difference between the drought magnitude DM and recovery threshold RT; the magnitude of $RT$ ($Z_{anom_t}$) can be small or large, as long as it is > ($Z_{LTA_m}$). I doubt if it is of any use for water management in the way it is defined now;
- Pg. 7-11, Chapter 4: do not use "central" England (e.g. pg. 9, lines 1 and 9). This region has not been identified Figs. 1 and 3. It is confusing. Stick to regions defined;
- Pg. 7, lines 9-10: I suggest to remove "…., and possibly 2003-2004.". This is not convincing in Northern UK;
- Pg. 7, line 11: you would expect (hard to read details in Fig. 3) ".., 2003-2007 and" ".., 2004-2007 and" because 2003 is a well-known drought year. You also mention it in line 10;
- Pg. 7, line 16: delete "…1943-1945…."; too little gauging stations with flow data to conclude this;
- Pg. 7, lines 22-23: "…followed by a long drought termination phase for catchments in South-west UK, whereas…". I believe this also applies to an equal number of catchments in Anglian;
- Pg. 7, line 25 and pg. 8, line 4: it reads as contradiction. "1995-1998 was relatively coherent at a regional scale" (pg. 7), whereas it is classified as "…the most nationally coherent event …" (pg. 8);
- Pg. 8, line 7: Can you add the catchment in drought in "….but one of the study catchments (Fig. 4; left), offering…". Is it the Tay (ES)?;
- Pg. 8, line 9: you mean that the drought duration is 3 years, but the text suggest that the drought in de south is 3 year longer;

- Pg. 8, lines 14-16: what about the exceptions in these regions (at least 3 catchments);
- Pg. 8, line 18 and pg. 9, line 10: I would not explicitly mention the Thames. The catchment belongs to SE;
- Pg. 8, line 19: add reference(s) "….referred to as the 1995–1997 drought in the literature, it was…";
- Pg. 8, lines 26-27 and pg. 9, lines 1-2: nothing to say about catchments with > 3-seasons drought terminations?;
- Pg. 9, lines 10-11: description of start and end of drought termination implies Win-Spr or Win-Sum. I do not believe this is consistent what is said on pg. 9, lines 26-28 and pg. 10, lines 1-2.
- Pg. 9, lines 24-25: can you say that a "rate" is abrupt? A termination can be abrupt;
- Pg. 10, lines 16-17: revise "…and two drought characteristics (drought magnitude and duration of drought development) were…" into "…and two drought development characteristics (drought magnitude and duration) were…";
- Pg. 10, lines 24-26: is this not obvious because *n=459* (all individual events) instead of *n=52* (catchment-average)?;
- Pg. 11, lines 9-10: I suggest to indicate in Table A.1 which catchments are included in the subset (17 out of 52);
- Pg. 12, line 5: revise "…Two aspects were explored, …" in "…Two aspects were explored in the next section, …";
- Pg. 12-16. Discussion: I would expect here (results from pg. 7, Section 4.1) a comparison with literature dealing with spatial distribution of drought in the UK (e.g. Hannaford et al., 2011; deals in Fig. 4 with spatial extent of catchments in 4 GB regions that experience abnormal low flows; in total over 100 catchments);
- Pg. 13, line 23-24: You have defined drought magnitude DM as the largest negative $Z_{anomt}$ value. DM might have a too instantaneous nature rather than a integrative nature, which could explain the observed weak correlation with drought termination duration;
- Pg. 14, line 10: revise "…than any other event since 1929, …" in "…than any other event since flow gauging started in 1929, …";
- Pg. 14, lines 20-23: I believe that Fig. 6 is not required. You can simply refer to Marsh et al. (2013), if it is just a copy. If not then you need to say what you have changed / derived from the data / results;
- Pg. 16, lines 5-7: I trust that you can refer to Fig. 5, seasonality map (bottom right);
- Pg. 16, lines 27-30 and pg. 17, lines 1-2: does this mean that the drought termination seasonality at the UK national scale is likely to be influenced by one of a combination of the three mentioned large-scale drivers? If so, it should be elaborated, if not, then I suggest to leave out these sentences;
- Pg. 17, lines 12-22: impact of selected parameters on type of droughts (i.e. multi-year droughts) and under representation of short-lived drought, responsive catchments should not be in Conclusions, but in Discussion;
- Pg. 17, lines 23-26, pg. 18, lines 1-2: influence of monthly time step should not be in Conclusions, but in Discussion;
- Pg. 18, lines 3-10: description of influence of abstractions that drought-terminating rainfall must account for this "anthropogenic deficit" in addition to the natural river flow deficiencies, should not be in Conclusions, but in Discussion;
- Pg. 19, lines 2-4: you are right about lack of groundwater chronologies, but do these not exist for other hydrometeorological variables? You refer to number of papers that deal with these (e.g. Dettinger, 2013; Mo, 2011);
- Pg. 24, caption: Duplication with caption Fig. 2 "Drought termination characteristics denoted as follows: DTD = drought termination duration; DTR = drought termination rate. Drought

development characteristics are denoted as follows: DDD = drought development duration; DM = drought magnitude." Only needed in one caption;
- Pg. 25, Table 2: revise headings and order as follows:

| Catchment | Number of drought events | Drought termination rate (%.month$^{-1}$) | | Year of drought Termination ranking 2$^{nd}$ by drought termination rate |
| --- | --- | --- | --- | --- |
| | | 2009-2012 | Rank 2 | |
| Severn | 16 | 90.6 | 26.5 | 1997 |
| | | | | |

"Rank (out of total number)" can be replaced with "Number of drought events". Easier to understand.
- Pg. 28, Fig. 1: Add acronyms and description of regions from caption Fig. 3 in caption Fig. 1. Add acronyms also in top left legend. Do not use colours in inset (upper right). These colours are confusing because there is no link with the colours in map with catchments;
- Pg. 29, Fig. 2: you did not define +ve and –ve. It is $Z_{anom\,t}$;
- Pg. 29, caption Fig. 2: revise "…within D; and T is the number of months of above average flows required for the end of the drought termination phase." in "…within D; and T is the number of consecutive months of above average flows required for the end of the drought termination phase.";
- Pg. 30, Fig. 3: How to interpret the width of the red and blue bars? Add temporal resolution of bars. Is a decade (e.g. 1970-1980) subdivided in 120 monthly time steps (x-axis);
- Pg. 31, Fig. 4 and pg. 32, Fig. 5: legend at top "Duration (months)". I believe that a duration of 1 month is impossible when T is set at 2 months;
- Pg. 31 and pg. 32, captions Fig. 4 and 5: acronyms and description of regions from caption Fig. 3 should go to caption of Fig. 1, and can be deleted here (duplication);
- Pg. 33, Fig. 6: not needed (see remark pg. 14, lines 20-23).

Hannaford  et al. (2011): Examining the large-scale spatial coherence of European drought using regional indicators of precipitation and streamflow deficit. Hydrol. Process. 25, 1146–1162.

---

## Author Comment (AC1) · 8 Apr 2016

General Comments

The reviewer raises important points on the novelty of the approach and the overgeneralisation of results which we have addressed below. We hope that our responses are acceptable, and we thank the reviewer for their constructive and complimentary review which has improved the manuscript.

Specific Comments

Page 4 Lines 3–10:

Whilst some of the studies cited conduct analysis at the monthly time step (as we do here), and some use the PDSI which accounts for the water balance over recent

months, these studies do refer to a day (e.g. Kam et al. 2013) or month (e.g. Patterson et al. 2013) in which drought termination occurs. Where SPI3 (e.g. Kam et al. 2013) or a three-month termination criterion (e.g. Patterson et al. 2013) are used, any implied 'termination duration' is 'hard coded' to be three months. This does not give an appreciation of the variability in the duration of drought termination, and in both studies the day or month of drought termination (as an instantaneous point in time) is further analysed (e.g. for the season in which that point in time occurs). Our study differs from these because drought termination is a defined period of a drought event with its own start and end and a duration in between these points. The drought termination rate is the magnitude of change in river flow anomalies over time during this period and the seasonality is the seasons through which the period occurs. Our approach could complement existing threshold-based methods by subdividing an identified period of drought into drought development and drought termination phases based on the minimum value of the index used (e.g. PDSI). We have updated the text at the end of the Introduction to clarify the differences between our approach and those of other studies.

Page6, Lines 18–23:

We thank the reviewer for their comments on this important aspect of our approach. The decisions on the parameter values are probably the most important factor in the number and characteristics of the identified drought termination events. For a previous application (not published), we conducted a very preliminary sensitivity analysis which demonstrated the impact of varying the parameter values. For this application, we tested a smaller number of combinations of parameter values (informed by that previous sensitivity analysis) and found that values of 10, 1 and 2 for D, R and T (respectively) identify droughts (and terminations) that are well documented in the literature (e.g. Marsh et al. 2007 and Parry et al. 2013, both cited in the manuscript). These values also capture the spatial variability in drought risk in the UK (lower in the north and west, higher in the south and east). The reviewer is correct that there are instances in the chronologies of drought termination presented in the manuscript when

a drought termination period is immediately followed by the next drought development phase (when two months are above average followed by nine out of the next ten months below average) which would be classified as the same event if T=3. However, the same issue would arise if T=2,3,4,…... The subjective decisions we have made here are not different to those of many studies in the literature on threshold-based drought indices which may make arbitrary choices on the threshold quantile and n-month accumulation periods. We agree with the reviewer that a comprehensive sensitivity analysis is required, but it is a complex question that we believe is worthy of a study in its own right. This paper aims to be a proof of concept that the approach is useful in systematically identifying and characterising drought terminations in the historical record. We have strengthened the text in the discussion to explain our future plans to more comprehensively address the question of parameter selection.

Page 6 Line 27:

We have tested for normality in each of the series used in correlation analysis through the Shapiro-Wilk test and quantile-quantile (Q-Q) plots. The majority of the series are not normally distributed so the use of Spearman correlations is justified. We have modified the manuscript to better justify our use of the Spearman approach.

Page 4 Line 23:

We have restructured the first sentence to emphasise the selection criteria and de-emphasise the importance of the number of catchments which satisfy these criteria.

Page 10 section 4.4:

The sentence relates specifically to the 1995-1998 and 2009-2012 events, drawing on the analyses provided in sections 4.2 and 4.3. This statement also holds true for some other events; for example, the top 5 drought termination durations for the 1973 event are the Bedford Ouse, Wensum, Lud, Stringside and Colne (see Figure 3), all of which are in Anglian region and have moderate to high BFI values (0.52-0.90), indicative

of groundwater influence in the catchments. However, we agree with the reviewer that the link between larger groundwater influence in catchments and longer drought termination durations does not apply for all identified drought termination events. We have extended the sentence to acknowledge that this does not apply to all events.

Similarly, on Page 12 Lines 11–16:

We accept that the spatio-temporal distribution of rainfall will impact the spatio-temporal variability of drought termination in river flows. Two of the most important factors in the characteristics of drought termination in a given catchment are the amount and timing of rainfall and the modulating effect of the catchment characteristics. We only claim on page 12, lines 11-16 that characteristics are partly (i.e. not wholly) attributable to catchment characteristics, but we agree that the link to rainfall could be made more explicitly. We have modified the manuscript accordingly.

Technical Corrections

Abstract, Line 1:

We have removed "storms and" from the first line of the Abstract.

Page 15 Lines 3:

We have replaced the numbers with percentages, as well as the reference to "five catchments" later in the same sentence (for consistency).

---

## Author Comment (AC2) · 8 Apr 2016

General Comments

We thank the reviewer for their very comprehensive review and positive conclusion. The comments provided by the reviewer are constructive in their nature and have helped to considerably improve the manuscript. We have responded below to each of the points in turn, providing the clarifications requested and making the changes necessary. We hope that the reviewer finds our responses acceptable so that we can revise and improve the manuscript accordingly.

Major Comments

1) We thank the reviewer for their thoughts on the methodological approach applied

in this study. Using the drought magnitude (DM) to subdivide a drought into drought development and drought termination is a core element of our approach. The decision that the DM should be the maximum negative anomaly (rather than the absolute lowest flow) was taken to objectively compare droughts and drought terminations that occur in different seasons. We agree wholeheartedly that the decisions on the parameter values are probably the most important factor in the number and characteristics of the identified drought termination events. This was demonstrated by a very preliminary sensitivity analysis as part of a previous application (not published). Following this test case, we realised that this is a complex topic and worthy of a more comprehensive analysis that we believe is beyond the scope of this already relatively long paper. For this application, we tested a number of different combinations of parameter values (informed by that previous sensitivity analysis) and decided upon 10, 1 and 2 for D, R and T (respectively) because they identify droughts (and terminations) that are well known and which appear in the literature (e.g. Marsh et al. 2007 and Parry et al. 2013, both cited in the manuscript). As we suggest in the manuscript, these parameters identify multi-year to multi-season droughts well and capture the spatial variability in drought risk (lower in the north and west, higher in the south and east of the UK). Our study is one of many in the literature that must make subjective decisions on parameter values related to threshold-based drought indices, such as the threshold quantile and any n-month accumulation period. One of the main aims of this paper is a proof of concept to demonstrate the utility of the approach in systematically identifying and characterising drought termination in the historical record. The next stage will be to undertake a robust assessment of the sensitivity of the results to parameter values to provide advice to users. This is now included in the discussion.

2) We recognise that many drought studies apply a lower threshold than the average monthly flow such as Q70, Q80, Q90 or Q95. We have not applied any of these lower thresholds but it can be assumed that the durations of drought overall (and therefore both drought development and drought termination phases) would decrease. A lower threshold is likely to sub-divide long duration events into a number of shorter more extreme episodes each of which would have a drought termination phase. It is difficult to envisage a well constrained drought (e.g. 2010-12) containing n Q80-derived droughts, for example, each with their own termination. In order to focus on the multi-season to multi-year events (pg. 6, line 20) which cause water supply problems, a duration-based approach using a higher threshold is required. The question of the most appropriate threshold will also be subject to a sensitivity analysis, but is outside the scope of this paper as a proof of concept. We acknowledge that the suitability of a given threshold differs depending on individual perceptions or applications and have added text in the discussion to provide this caveat.

3) We agree with the reviewer that deficit volume based approaches are certainly important for some studies on the recovery from drought, such as to replenish stores within the catchment (e.g. reservoirs or aquifers). However, river flows are naturally integrative and the focus of this study is on river flow dynamics rather than recovering a volume of water in a river that was 'lost' during drought development. We have included text in the discussion section to reflect these different approaches.

4) One of the main overall aims of the study is "assessing the full range of drought termination types and characteristics" (pg.3, lines 14-15). The two brief case study events (1995-98 and 2009-12; sections 4.2 and 4.3) were chosen to provide a contrast between a more gradual event (1995-98) and a more abrupt event (2009-12). We recognise that the focus on 2009-12 in sections 5.2 and 5.3 may shift the focus towards abrupt events, but this was only to put the most recent event in its historical context (we could have performed the same analysis of historical context on the more gradual 1995-98, for example). We identified three comparably abrupt events to 2009-12 for the Thames catchment, but we do not say that these are the only abrupt events (4/35) for this catchment. The 2009-12 event was an extreme drought termination event; it is likely that smaller values of DTR also caused substantial problems for water managers.

5) We agree with the reviewer and have decided to remove this sentence from the manuscript.

[Figure]

6) We accept that the correlations presented in the manuscript are relatively weak and cannot yet be the basis of water management decisions. We have removed the suggested sentence and caveated a corresponding part of the conclusion (pg. 18, lines 20-23).

7) Whilst the approach used in the manuscript could be applied to groundwater level data, we stand by our view that this would be beyond the scope of the study which was to demonstrate that the concept can be used to systematically analyse hydrological drought termination. Future work will provide a similar systematic assessment of drought termination in long groundwater level records and show comparisons with those derived from river flows to better understand the complex concept of the propagation of drought termination. The reviewer is correct that drought termination in river flows may not correspond to drought termination in the associated groundwater level records, but this does not necessarily imply that river flow terminations have been incorrectly identified. There are also important differences between drought terminations identified in river flow and groundwater level records even within the same catchment; boreholes provide an understanding of a very localised part of a heterogeneous aquifer whereas river flow records integrate over a larger area. This question of the propagation of drought termination through the hydrological cycle is a key question that our approach could address but we feel is a large enough topic for a study in its own right.

8) We have moved the paragraphs mentioned in the reviewer comments into the discussion section.

9) It is true that the termination is traditionally an instantaneous point in time. However, we feel that 'recovery' is also a loaded term that may create confusion amongst readers. Recovery is frequently used in ecological studies to refer to the resilience of ecosystems and can relate to a period of up to five years or more over which plants and animals return following a drought disturbance. In hydrology, recovery might imply the longer-term cumulative water deficit method which our approach does not use (see response to point 3 above).

Minor comments

Pg. 2, lines 1-2: By removing the specific reference to recent events in the UK and combining the first two sentences of the abstract, we have reduced the emphasis on abrupt terminations and strengthened the recognition that there are a wide range of possible scenarios for drought termination.

Pg. 2, lines 21-24: We have removed the element of the sentence that implies potential use for water resources management (given the lack of strong relationships) and we have been more specific about the direction of correlations.

Pg. 3, lines 1-3: We have reduced the length of this sentence.

Pg. 3, lines 1-14: We have removed references to violent weather and flooding, and re-structured the first paragraph to better reflect the range of possible scenarios of drought termination.

Pg. 3, lines 20-26: We have made reference to these two papers in terms of their consideration of the end of a drought.

Pg. 4, line 6: Following our response to point 9 above, we hope to maintain the ter-minology that is used consistently throughout the manuscript: drought termination as a phase of drought. We have modified the sentence explaining Bonsal et al. (2011) and Nkemdirim & Weber (1999) to better explain that these two studies also apply the concept of drought termination as a phase.

Pg. 4, line 20: We have rephrased this sentence so that representativeness is not concluded from having coverage of ∼40% of the gauged area.

Pg. 5, lines 26-27 and pg. 6, lines 1-3: We have added text into the anthropogenic influences paragraph of the discussion to recognise the north-west / south-east bias.

Pg. 5, line 22: We have now included text that recognises the shorter records and ex-plains the calculation when this applies. We have included statistics on data availability

for those shorter records that intends to reassure the reader that the LTA values are derived from large enough sample sizes of data.

Pg. 6, line 6: We agree with the reviewer that the use of the term 'threshold' is confusing to readers. We have renamed this as the 'termination magnitude' (or TM) and have revised Fig. 2 accordingly.

Pg. 6, line 12: The DTR provides an indication of the slope of a line from the DM to the RT (now TM). We agree that the RT (TM) is an arbitrary point and we could use instead the average over the two months of >ZLTAm, for example. We believe that the DTR is potentially useful to water managers. For two events of the same duration, a higher DTR indicates a more rapid transition from drought to potential flooding. The research presented in this manuscript focuses on the identification of events and their characterisation (including their DTR). Water managers may use the information provided by the historical chronologies to better understand how different types of catchments respond to different scenarios, and could tailor decisions or actions accordingly.

Pg. 7-11, Chapter 4: We have updated references to "central" England to now read "the Midlands" because these terms essentially refer to the same geographic region.

Pg. 7, lines 9-10: On reflection we agree with the comments of the reviewer so we have removed this event.

Pg. 7, line 11: The use of both "2003-04" and "2004-07" is deliberate in order to differentiate between two different events. For many catchments in south-eastern England, the events are identified separately. The 2003 event was not as severe in the UK as in Europe, and the 2004-07 event was much more problematic than the 2003 event in south-eastern England. Hopefully now that "2003-04" has been removed (see response above) any confusion can be avoided.

Pg. 7, line 16: On reflection we agree with the comments of the reviewer so we have removed this event.

Pg. 7, lines 22-23: We agree with the comments of the reviewer and have added Anglian into this statement.

Pg. 7, line 25 and pg. 8, line 4: We agree that this reads as a contradiction so we have removed the first statement and clarified the second statement.

Pg. 8, line 7: We propose to retain the current text because we prefer the reader to consider the widespread nature of drought rather than being pre-occupied with why a specific catchment was the exception.

Pg. 8, line 9: The reviewer is correct that we mean a three-year overall drought duration in the south and east, so we have clarified the text accordingly.

Pg. 8, lines 14-16: We have specified the exceptions to this statement.

Pg. 8, line 18 and pg. 9, line 10: We have removed the two references to Thames region.

Pg. 8, line 19: We have added two example references from the literature.

Pg. 8, lines 26-27 and pg. 9, lines 1-2: We have added text to acknowledge the prevalence of >3-season drought terminations.

Pg. 9, lines 10-11: We agree with the reviewer that there is inconsistency and we have modified both paragraphs for clarity.

Pg. 9, lines 24-25: We agree with the comments of the reviewer and have made this modification.

Pg. 10, lines 16-17: We have made this modification.

Pg. 10, lines 24-26: We have removed this sentence.

Pg. 11, lines 9-10: We have indicated in Table A1 the catchments which are included in the subset, and have referred to Table A1 on pg.11, lines 9-10.

Pg. 12, line 5: We have made this modification.

Pg. 12-16. Discussion: We have added a new section into the discussion which evaluates the chronologies of drought termination relative to the wider literature on the spatio-temporal distribution of drought (and drought termination) in the UK.

Pg. 13, line 23-24: Even though the DM is an instantaneous value, one would think a larger DM is more likely to lead to longer drought termination duration (DTD), as found by Nkemdirim & Weber (1999), rather than shorter. For responsive catchments, it may be that DTD is insensitive to DM because the rainfall input dominates the trajectory of drought termination.

Pg. 14, line 10: We have made this modification.

Pg. 14, lines 20-23: We have removed Fig. 6 and now refer to Marsh et al. (2013).

Pg. 16, lines 5-7: We have added a reference to Fig. 5 bottom right.

Pg. 16, lines 27-30 and pg. 17, lines 1-2: We have modified the text in order to retain some of the different synoptic drivers that have been shown to be influential on drought termination, but to clarify that further work is required to assess whether these factors are important in the historical chronology of drought termination for the UK.

Pg. 17, lines 12-22: We have moved this section into the discussion.

Pg. 17, lines 23-26, pg. 18, lines 1-2: We have moved this section into the discussion.

Pg. 18, lines 3-10: We have moved this section into the discussion.

Pg. 19, lines 2-4: We have removed the reference to hydrometeorological variables but maintained groundwater and merged it with the following sentence on water quality and ecology.

Pg. 24, caption: There is no duplication in explaining the definition of DTD, DTR, DDD and DM between the captions of Table 1 and Fig. 2. The terms DTD, DTR, DDD and DM are not used in the caption of Fig. 2.

Pg. 25, Table 2: We have revised the table headings in Table 2 as suggested by the reviewer.

Pg. 28, Fig. 1: We have added acronyms and description of regions from the caption of Fig. 3 to the caption of Fig. 1 (and removed them from the caption of Fig. 3 to avoid duplication). We have added acronyms to the legend in the top left. We have removed the colours from the inset map and now use lines to label constituent countries.

Pg. 29, Fig. 2: We have removed the '+ve' and '-ve' directions for Zanom.

Pg. 29, caption Fig. 2: We have added the word "consecutive" into the caption of Fig. 2.

Pg. 30, Fig. 3: We have modified the caption of Fig. 3 to indicate that a decade can be subdivided into 120 monthly time steps as well as information on the total number of time steps along the x-axis. We have deleted the acronyms denoting the regions from the caption of Fig. 3, moving them instead to the caption (and legend) of Fig. 1.

Pg. 31, Fig. 4 and pg. 32, Fig. 5: We have modified the duration legend accordingly.

Pg. 31 and pg. 32, captions Fig. 4 and 5: We have deleted the acronyms and descriptions of regions from the captions of Fig. 4 and Fig. 5 and added them into the caption of Fig. 1.

Pg. 33, Fig. 6: We have removed Fig. 6.

---

## Referee Report (RR1)

**A systematic assessment of drought termination in the United Kingdom**

S. Parry, R.L. Wilby, C. Prudhomme and P.J. Wood

*Hydrol. Earth Syst. Sci. Discuss., revised manuscript 26 May 2016*

The authors substantially revised the first manuscript based upon the comments of the reviewer. The study is original and it certainly is worthwhile to be published in HESS, after responding to a few (minor) comments.

I understand your point that a full sensitivity analysis would imply another comprehensive study, which is beyond the scope of this paper. I appreciate that you added a separate section on the sensitivity of the parameters to identify drought chronologies (Section 3.2), which refers to Fig. 3 that shows the sensitivity for 2 contrasting UK catchments. I think that this limited sensitivity analysis in principle is sufficient to make readers aware that the presented chronologies depend on the choices of the parameters, in particular *D*, *R*, and *T*. However, the sensitivity aspect is not mentioned in the Abstract and Conclusions. I believe that one sentence about the sensitivity needs to be added in both sections.

Through the parameter selection (*D*=10, *R*=1, *T*=2), focus is on the termination of multi-year, multi-season drought events. This also needs to be mentioned in the Abstract and Conclusions. For suggestions, see minor comments below.

In the Discussion section you mention that the selected monthly average flow threshold is higher than those sometimes (HvL: usually) applied in threshold-based studies. This requires a justification. I suggest to include the justification made by you in reply to my comment 2 (Hydrol. Earth Syst. Sci. Discuss., doi:10.5194/hess-2015-476-AC2, 2016).

I welcome the paragraph in the Discussion about the importance of missing water (deficit volume, e.g. for reservoir of aquifer management), which is very relevant for the managers of these resources.

As said earlier, this study is one of the first on ending of a drought. It would be good to set the terminology right from the beginning. The authors use the term "termination" for the last phase of a drought (decrease of the deficit to zero), whereas I thought that termination is more associated with an instantaneous point in time. I would the use termination for the latter and "recovery" for the last phase. The authors come in their reply with a good number of reasons (e.g. from ecology) to use the terms in the manuscript. They clarified this in the text. I believe we need to find another platform to discuss terms associated with the ending of a drought.

Minor comments:
- Pg. 1, line 19: Add/revise "…..regionally-coherent, termination of multi-year, multi-season drought events……";
- Pg. 3, line 12: you can add here a line about the sensitivity analysis and that the study aims at detection of the termination of multi-year, multi-season drought events;
- Pg. 4, line 16: something is wrong with the format of the Equation; does not properly show up in the document;
- Pg. 5, lines 18-23: *D* and *R* are explained, whereas this is missing for *T*=2;

- Pg. 13, lines 9-23: The heading of Section 5.2 is about drought "termination" whereas the text section is about entire droughts. Could you make the validation of the termination a bit more specific?
- Pg. 17, line 22: Add/revise "For the first time, termination of multi-year, multi-season drought events have been......";
- Pg. 32, line 7: Add: "...rate. $D7$-$D36$: 7-36 months that for which $Z_{anom}$ is negative, $R1$-$R3$: 1-3 months within the $D$-months duration for which $Z_{anom}$ is permitted to be positive, $T2$-$T6$: 2-6 consecutive months for which $Z_{anom}$ is positive. The bold box ...".

---

## Author Response (AR2)

**Covering letter: Author response to Editor Report**

Dear Lena

Thank you once again for reviewing the latest version of our manuscript. We are glad that the most substantial change relating to sensitivity analysis of parameter values has improved the paper.

We have addressed the comments suggested by both the reviewer and editor, all of which help to further improve the manuscript. Please find the revisions in the marked-up version below and the revised manuscript attached.

If we can provide any further clarification or modifications, please let me know.

Best wishes,

Simon Parry

**Author Comment in response to review by Dr. van Lanen**

We thank the reviewer once again for their very helpful suggestions on how to further improve the manuscript. We are glad that we have mostly addressed the points identified by the reviewer in our initial response and modified manuscript, and we welcome the opportunity to respond to these additional comments.

In order to address comments on the need to emphasise the multi-season to multi-year focus of our study, we have added sentences to the Abstract, Introduction and Conclusion to underline this point. We have also added text to these same three sections on the presence of a sensitivity analysis in the study. In most cases, we have added the multi-year focus and sensitivity analysis together in the same sentence, because they are related to one another.

We have enhanced the discussion around the threshold level used in our study relative to those used by others, as well as the impact this would be expected to have on derived drought characteristics. As suggested by the reviewer, we used our previous author response as the basis for these modifications.

We have addressed all of the minor comments proposed by the reviewer. The most substantial of these changes referred to the need to focus more on validation of drought termination in section 5.2. We have added a new paragraph relating specifically to drought termination and included three references to show that our chronologies are corroborated by existing literature, whilst acknowledging that this is more difficult for drought termination due to the relative lack of focus within drought research.

**Author Comment in response to Editor Report**

We thank the editor for their feedback on the revised manuscript and we are satisfied that we have adequately addressed the majority of the comments provided by the reviewers. We are thankful for the opportunity to further improve the manuscript by addressing the additional suggestions of the editor.

We agree with both the reviewer and editor that it is important to underline the importance of the multi-season / multi-year focus of our study. To address this suggestion, we have added sentences to the Abstract, Introduction and Conclusion. We hope that this key element of our work is now much clearer to the reader.

We have added text to the discussion section around the selection of threshold levels, including a few references on the use of different criteria in existing studies.

We have modified the beginning of section 5.1 to ensure the phrasing is less vague. We have more clearly described the balance between rainfall and catchment characteristics, and added two more sentences at the end of the paragraph to describe how this might be expected to vary spatially in the UK.

We thank the editor for highlighting a number of more minor corrections that are required, and have addressed all of them.

In section 4.4, our definition of statistical significance is now at the beginning and we have removed words like 'strong' and 'weak' where possible. Generally speaking, most of the correlations in the first paragraph are significant and most in the second paragraph (the subset) are not. Also, within each of the paragraphs the significant correlations proceed the insignificant correlations. The final paragraph relates to correlations between drought development and drought termination, which we believe should follow both of the paragraphs on correlations between drought termination and catchment characteristics.

Finally, we thank the editor for the very helpful reference that was recommended. We agree that this reference is relevant for our study and have cited this in both the parameter sensitivity and methodological discussion sections.

[revised manuscript text omitted]